# Monoallelic expression and epigenetic inheritance sustained by a *Trypanosoma brucei* variant surface glycoprotein exclusion complex

Joana Faria[1], Lucy Glover[1,2], Sebastian Hutchinson [1,3], Cordula Boehm[1], Mark C. Field [1] & David Horn [1]

The largest gene families in eukaryotes are subject to allelic exclusion, but mechanisms underpinning single allele selection and inheritance remain unclear. Here, we describe a protein complex sustaining variant surface glycoprotein (*VSG*) allelic exclusion and antigenic variation in *Trypanosoma brucei* parasites. The *VSG*-exclusion-1 (VEX1) protein binds both telomeric *VSG*-associated chromatin and VEX2, an ortholog of nonsense-mediated-decay helicase, UPF1. VEX1 and VEX2 assemble in an RNA polymerase-I transcription-dependent manner and sustain the active, subtelomeric *VSG*-associated transcription compartment. *VSG* transcripts and VSG coats become highly heterogeneous when VEX proteins are depleted. Further, the DNA replication-associated chromatin assembly factor, CAF-1, binds to and specifically maintains VEX1 compartmentalisation following DNA replication. Thus, the VEX-complex controls *VSG*-exclusion, while CAF-1 sustains VEX-complex inheritance in association with the active-*VSG*. Notably, the VEX2-orthologue and CAF-1 in mammals are also implicated in exclusion and inheritance functions. In trypanosomes, these factors sustain a highly effective and paradigmatic immune evasion strategy.

---

[1] The Wellcome Trust Centre for Anti-Infectives Research, School of Life Sciences, University of Dundee, Dow Street, Dundee DD1 5EH, UK. [2] Present address: Trypanosome Molecular Biology, Department of Parasites and Insect Vectors, Institut Pasteur, 25-28 Rue du Docteur Roux, 75015 Paris, France. [3] Present address: Trypanosome Cell Biology Unit, INSERM U1201, Department of Parasites and Insect Vectors, Institut Pasteur, 25-28 Rue du Docteur Roux, 75015 Paris, France. Correspondence and requests for materials should be addressed to D.H. (email: d.horn@dundee.ac.uk)

Antigenic variation and host immune evasion by parasites causing malaria[1], giardiasis[2] and African trypanoso-miasis[3] depend upon monoallelic and switchable variant surface antigen expression. Indeed, allelic exclusion governs the expression of many of the largest known gene-families, typically encoding cell-surface proteins in protozoa and mammals, and has major impacts on health and disease. In mammals, odour detection depends upon olfactory receptor exclusion[4] and B- and T-cell specificity depends upon immunoglobulin and surface receptor exclusion, respectively[5]. Selection of a single allele from a large gene family is thought to be inherently stochastic rather than deterministic and generates tremendous cellular diversity, upon which adaptive mechanisms operate. Notably, the diversity-generating exclusion systems of protozoan antigenic variation, and the host immune response, are mutually adaptive and in direct competition. Despite intense study, the molecular mechanisms facilitating the selection and maintenance of a single active allele, coordinated with heritable exclusion of all others, remain poorly understood.

The African trypanosome, *Trypanosoma brucei*, is transmitted among mammalian hosts by tsetse-flies and, due to effective immune evasion, causes chronic and lethal infections, specifically sleeping-sickness in humans and nagana in cattle. In these parasites RNA polymerase-I (pol-I) transcribes a single, domi-nant, telomeric variant surface glycoprotein (*VSG*) gene[3] as part of a polycistronic transcription unit. Pol-I transcription initiates at several sites but this is attenuated at all but one site[6]. *T. brucei* can switch to activate a new telomeric *VSG* expression site (ESs) but, significantly, switching occurs at low-frequency and all but one of the ESs typically remain 'silent', despite possessing all *cis*-elements required for expression[7].

The active *VSG*-ES is associated with an extranucleolar pol-I transcription factory, known as the expression site body (ESB)[8,9]. This *VSG*-ES is depleted of nucleosomes[10,11] and nuclease-hypersensitive chromatin persists at this locus even if transcrip-tion is blocked[12]. An HMG chromatin protein is enriched at both the ESB and the nucleolus[12,13], and a focus of SUMOylation associates with the ESB[14]. In addition, histones[15] and histone variants[16], telomere-binding proteins, a histone H3-K76 methyl-transferase, a histone deacetylase, a chromatin remodeler, histone chaperones, nuclear lamina components, cohesin and the inositol phosphate pathway all impact *VSG*-ES silencing to varying degrees (reviewed in ref. [17]). These factors may impact chromatin-dependent silencing or transcriptional permissivity but, notably, a direct role in gene selection has not been demonstrated. More recently, the first factor specifically enriched in association with the active *VSG*-ES, VSG exclusion 1 (VEX1), was identified[17].

Here, we identify and characterise a VEX1–VEX2 complex. The complex assembles a sub-nuclear domain in a transcription-dependent manner and maintains *VSG* allelic exclusion by negatively controlling transcription of other telomeric *VSGs*. Inheritance of *VSG* exclusion requires maintenance of the VEX-complex during S-phase, which depends upon the conserved chromatin assembly factor, CAF-1.

## Results

**A subtelomere- and *VSG*-associated VEX-complex**. We previously described *T. brucei* VEX1 (Tb927.11.16920), the only known protein specifically enriched in association with the ESB[17]. VEX1, therefore, is at the heart of the nuclear subdomain meditating antigenic variation in trypanosomes, but VEX1 lacks orthologs in other cell types, it remained unclear why *VSG*-exclu-sion was only partially perturbed following VEX1-knockdown and it remained unclear whether other factors were involved in *VSG*-exclusion.

To identify VEX1 chromatin interactions, we affinity-purified VEX1$^{myc}$-associated chromatin and deep-sequenced the enriched DNA. Reads were accurately aligned to individual *VSG*-ESs by uniqueness filtering, using MapQ > 1[18]. An examination of protein coding sequences within the hemizygous subtelomeric *VSG*-ESs[7,19] revealed *VSG-2*, the active *VSG*, as the most enriched gene (Fig. 1a, Supplementary Data 1, sheet 1). We also observed particularly strong enrichment of the region downstream of this gene (Fig. 1a). This is consistent with a focus of VEX1 that is adjacent to, rather than coincident with, the extranucleolar focus of pol-I at the active *VSG*-ES[17]. Notably, DNA immediately downstream of several silent ES-associated *VSGs* was also enriched (Fig. 1a, Supplementary Fig. 1). This is also consistent with our model proposing that conserved *VSG*-associated sequences, including the telomeric repeats, participate in VEX1- and homology-dependent *VSG*-silencing[17]. Thus, chromatin interactions connect VEX1 to the conserved sequences associated with telomeric *VSGs*.

Next, we affinity-purified VEX1$^{GFP}$-associated proteins using cryomilling and high-affinity nanobodies. The procedure was carried out initially in insect-stage *T. brucei* (Supplementary Fig. 2a, b), for which protocols were established[20], and then in bloodstream form *T. brucei* (Fig. 1b). Quantitative proteomic analysis (Fig. 1b, Supplementary Fig. 2a, b, Supplementary Data 1, sheet 2) revealed tag-dependent enrichment of green flourescent protein (GFP) and the same set of five proteins in four independent experiments; VEX1 was enriched, as expected, but also Tb927.11.13380, an ortholog of the nonsense-mediated mRNA-decay ATP-dependent superfamily 1-type helicases, UPF1/SMG2/NAM7/Rent1[21], (Fig. 1c, Supplementary Fig. 2c–e), now designated VEX2 (predicted 224 kDa, 2026 residues). Trypanosomes encode two UPF1-related proteins (Supplementary Fig. 2c). The putative *T. brucei* canonical UPF1 has been studied but it remains unclear whether classical nonsense-mediated decay operates[22]. VEX2 contains a UPF1-related helicase domain towards the *C*-terminus, including the canonical motifs involved in ATP binding and hydrolysis and nucleic acid binding (Supplementary Fig. 2d). Structural prediction suggested the presence of an α-solenoid architecture at the *N*-terminus (Supplementary Fig. 2e), in place of the typical UPF2-interacting domain[23]; α-solenoids are frequently involved in protein–protein or protein–RNA interactions. The other three proteins that displayed tag-dependent enrichment were all three components of chromatin assembly factor 1 (CAF-1; CAF-1a, Tb927.8.3980; CAF-1b, Tb927.10.7050, CAF-1c, Tb927.11.4970; Fig. 1b, c, Supplementary Fig. 2a), the evolutionary conserved hetero-trimeric replication-associated histone chaperone[24]. Notably, no pol-I components were enriched.

Super-resolution fluorescence microscopy revealed that, like VEX1, VEX2 localised to a single major sub-nuclear focus (detected in ~80% of nuclei) that was replicated and segregated at the appropriate phases of the cell cycle (Fig. 1d). Consistent with the proteomics evidence, VEX2-foci co-localised with VEX1 foci (Fig. 1e, Supplementary Movie 1) and the extranucleolar, histone-depleted pol-I compartment (Fig. 1f). We conclude that the VEX1–VEX2-complex is associated with the single *VSG* tran-scription compartment.

**VEX-complex compartmentalisation is transcription-dependent**. VEX1 foci are disrupted in cells treated with transcription inhi-bitors, indicating that active transcription is required for assembly[17,25]. However, it was unclear from these studies whether VEX1 was degraded or redistributed throughout the nucleoplasm. Protein blotting indicated that neither VEX1 (Fig. 2a) nor VEX2 (Fig. 2b) were degraded following pol-I specific transcription

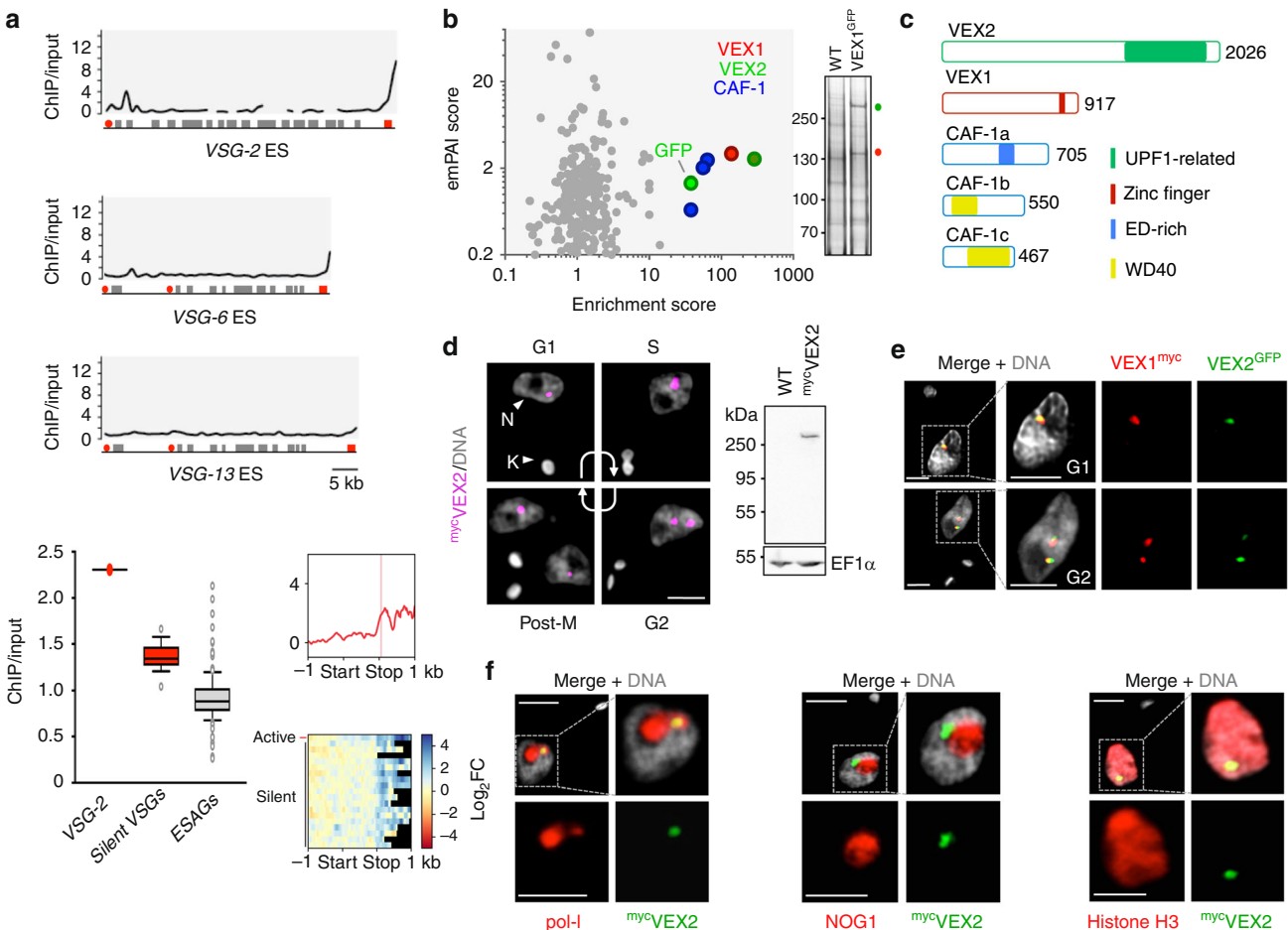

**Fig. 1** A subtelomere- and *VSG*-associated VEX-complex. **a** Affinity purification of VEX1[myc]-associated chromatin followed by sequencing (ChIP-seq). Enrichment traces over active (*VSG-2*) and silent (*VSG-6*, *VSG-13*) *VSG*-ESs with 1 kbp non-overlapping bins; red circles, promoters; red boxes, *VSGs*; grey boxes, expression-site associated genes (*ESAGs*). Box-plot depicting enrichment of the indicated CDSs. Centre lines show the medians; box limits indicate the 25th and 75th percentiles; whiskers extend from the 10th to the 90th percentile, outliers are shown. Silent *VSGs*, $n = 18$; *ESAGs*, $n = 129$. Metagene plot and the associated heat-map for active and silent telomeric *VSGs*. The red bar on the metagene plot indicates the location of a highly conserved sequence in the *VSG* 3′-untranslated region. **b** VEX1[GFP] immunoprecipitation and proteomics. GFP-tag dependent enrichment from bloodstream form *T. brucei*. emPAI, exponentially modified protein abundance index. The inset shows immunoprecipitates and tag-dependent VEX1 (red) and VEX2 (green) bands highlighted in a silver-stained gel. **c** Schematic representation of VEX1 (Tb927.11.16920), VEX2 (Tb927.11.13380) and CAF-1 subunits (a, Tb927.8.3980; b, Tb927.10.7050; c, Tb927.11.4970) indicating conserved domains and motifs. **d** Immunofluorescence analysis during the cell cycle and protein-blot analysis of [myc]VEX2 expression. N nucleus; K kinetoplast (mitochondrial genome). **e**, **f** Immunofluorescence-based colocalisation studies of VEX2[GFP] and VEX1[myc] (**e**) or a nucleolar (No) and ESB marker (pol-I, largest subunit), a nucleolar marker (NOG1) or Histone H3 (**f**). In **e**, G1 or G2 cells are shown. **d**–**f** DNA was counter-stained with DAPI; the images correspond to maximal 3D projections of 0.1 μm stacks; scale bars 2 μm; the data are representative of independent biological replicates and independent experiments. Source data are provided as a Source Data file

inhibition with BMH-21. Indeed, super-resolution confocal microscopy revealed that both VEX1 (Fig. 2a, Supplementary Fig. 3a) and VEX2 (Fig. 2b, Supplementary Fig. 3b) displayed significant levels of redistribution across the nucleus following inhibition of pol-I transcription, or inhibition of general transcription with actinomycin D (Supplementary Fig. 3c–f). Thus, recruitment of VEX1 and VEX2 to the *VSG* transcription compartment requires active pol-I transcription. VEX1 and VEX2 proteins remain within the nucleus when transcription is prevented, suggesting availability for retargeting once pol-I transcription reinitiates.

**VEX2 depletion results in multi-VSG expression.** VEX1 was initially identified from a high-throughput RNA interference (RNAi) screen for subtelomeric loss-of-silencing[17]. Another genome-wide RNAi screen indicated minimal fitness-cost

following VEX1-knockdown, but a significant and major fitness-cost following VEX2-knockdown, specifically in bloodstream-form cells and not insect-stage cells[26] (Supplementary Fig. 4a). This likely explains why only VEX1, and not VEX2, was identified in the loss-of-silencing screen, and indicates a bloodstream-form specific function for VEX2; VSG is not normally expressed in the insect-stage.

To explore the function of VEX2, we assembled three independent inducible RNAi knockdown strains with an active *VSG-2* expression site (Fig. 3, Supplementary Fig. 4b). Knock-down was associated with a significant growth defect (Fig. 3a), but little perturbation of cell cycle distribution (Supplementary Fig. 4c). In these cells, normally silent *VSG-6* was strongly derepressed after only 24 h, as assessed by protein immunoblotting (Fig. 3b), immunofluorescence microscopy (Fig. 3c) and immunostaining followed by flow cytometry (Fig. 3d). Notably, abundance of the initially active VSG was also clearly reduced

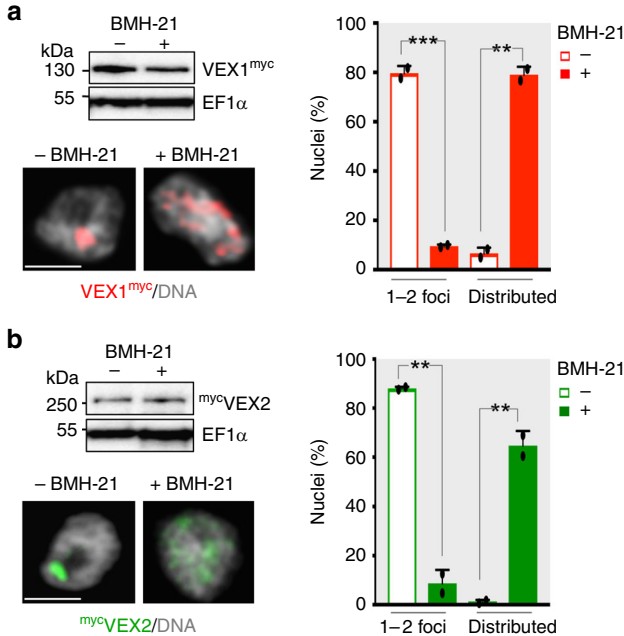

**Fig. 2** VEX-complex compartmentalisation is transcription dependent. **a**, **b** Protein-blot and immunofluorescence analysis of VEX1$^{myc}$ (**a**) and $^{myc}$VEX2 (**b**) before and after BMH-21 treatment (30 min at 1 μM). Proportions of nuclei displaying 1–2 major foci or distributed signals are indicated; the remaining cells displayed no detectable signal. Values are averages of two independent experiments (≥100 nuclei each). Error bars, SD; **p < 0.01; ***p < 0.001 (two-tailed paired Student's t test). DNA was counter-stained with DAPI; the images correspond to maximal 3D projections of 0.1 μm stacks; scale bars 2 μm; the data are representative of independent biological replicates and independent experiments. Source data are provided as a Source Data file

after 48 h of VEX2-knockdown (Fig. 3b) and VSG-6 appeared to accumulate in intracellular compartments as well as at the surface (Fig. 3c, right-hand side). To survey additional VSGs, we used quantitative proteomics of proteins released from the cell-surface. The active VSG on wild-type cells, VSG-2, displayed a relative abundance of 99.9%. Following VEX2-knockdown, VSG-2 remained the most abundant VSG, but VSG-6 and several additional ES-associated VSGs were derepressed (Fig. 3e, Supplementary Data 1, sheet 3). Because antigenic variation occurs at low frequency, *T. brucei* clones are typically homogeneous and express a single VSG (Fig. 3c–e). Following VEX2-knockdown, however, >90% of cells simultaneously expressed VSG-2, and VSG-6 at the cell surface as determined by microscopy (Fig. 3c) and flow cytometry (Fig. 3d). Taken together, these results indicate major disruption of allelic exclusion and the simultaneous expression of multiple VSGs on the surface of individual cells following VEX2-knockdown.

**VEX2 coordinates allele-specific *VSG-ES* transcription**. To further explore gene expression defects following VEX2-knockdown, we analysed the transcriptome, comparing three independent knockdown strains to three wild-type sub-clones (Supplementary Fig. 5, Supplementary Data 1, sheets 4–5). Since RNAi may not degrade the entire mRNA, we show efficient disruption of *VEX2* transcript in Fig. 4a. Among approximately 7600 genes encoded in the genome, 166 displayed >5-fold and significantly ($p < 10^{-2}$) increased expression after 24 h knockdown (Fig. 4b–e), including every known pol-I transcribed protein-coding locus. There are 18 known silent, telomeric, pol-I promoter-associated *VSGs* in the genome; 13 in polycistronic expression sites and 5 in monocistronic expression sites, with the latter only normally activated in metacyclic cells in the insect salivary gland[18]. Total transcript abundance from this full set of 18 *VSGs* (Fig. 4b, c), and 130 silent expression-site associated genes (*ESAGs*) (Fig. 4d), was increased by 212 and 32-fold,

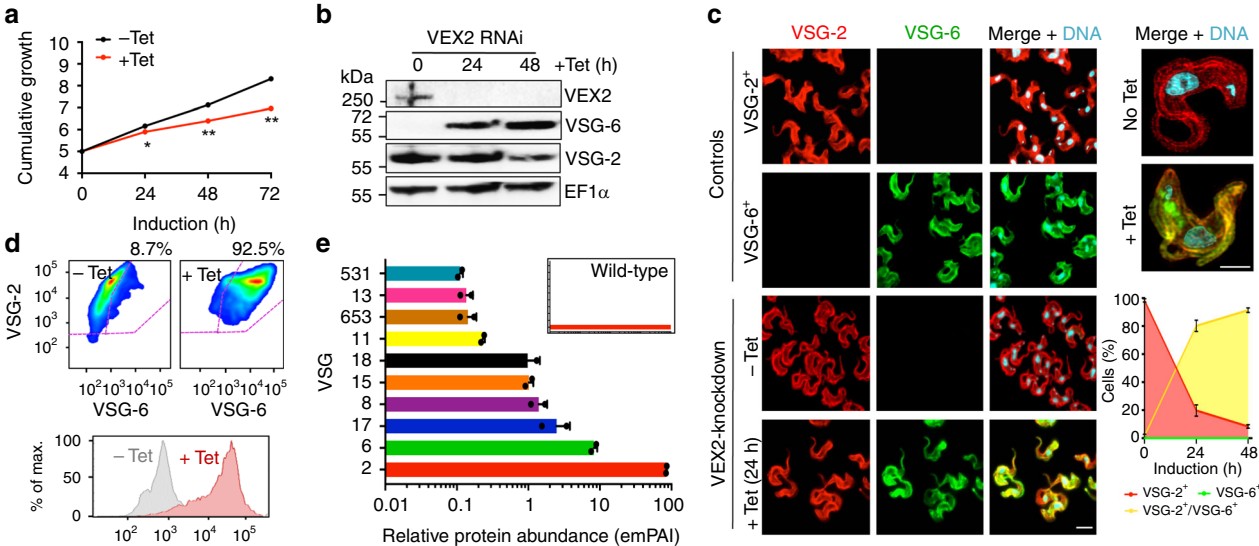

**Fig. 3** VEX2 depletion results in multi-VSG expression. **a** Cumulative growth following tetracycline (Tet) induced VEX2-knockdown; *p < 0.05; **p < 0.01 (multiple t tests). Protein blotting (**b**), immunofluorescence analysis (**c**), and flow cytometry (**d**), to assess *VSG* expression following VEX2-knockdown (24 h). EF1α, loading-control in (**b**); in **c**, DNA was counter-stained with DAPI and the graph depicts the rapid increase of dual VSG expressing cells following VEX2 knockdown by microscopy analysis. Scale bars, 5 μm (left hand side) and 2 μm (right hand side). Percentages in **d** indicate double-positive cells. **e** Quantitative mass spectrometry analysis of surface-VSGs following VEX2-knockdown (48 h). The inset shows data for wild-type cells. emPAI, exponentially modified Protein Abundance Index. Error bars (not visible in **a**), SD; data are averages from two (**a**/**c**/**e**) independent biological replicates and representative of independent experiments (**a**–**d**). Source data are provided as a Source Data file

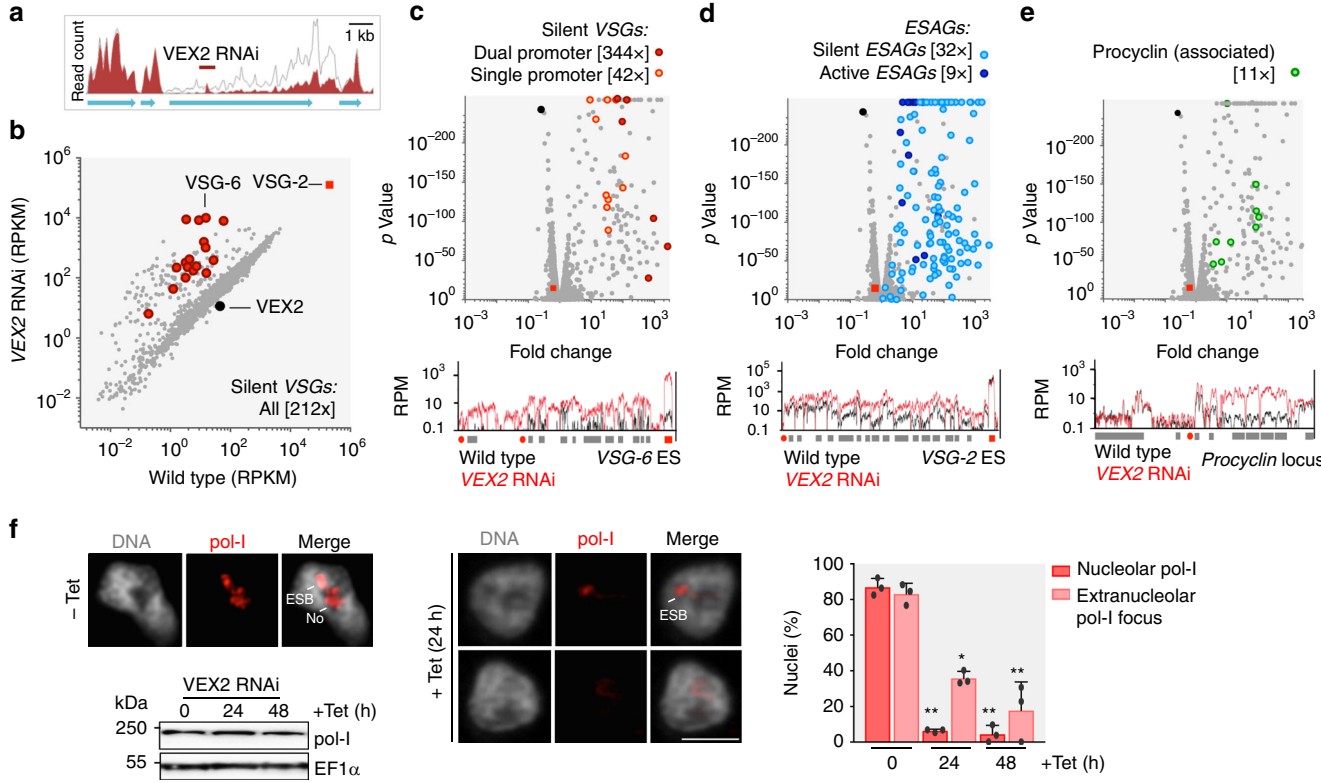

**Fig. 4** VEX2 coordinates allele-specific *VSG-ES* transcription. **a–e** RNA-seq analysis following VEX2-knockdown (24 h). Total number of reads at *VEX2* locus in the parental population or following *VEX2* RNAi (target fragment indicated) (**a**). **b–e** Values are averages from three independent RNAi strains relative to wild-type controls and numbers in brackets indicate increase in transcript abundance for each gene cohort. Red square, *VSG-2*; black circle, *VEX2*. Lower panels show transcript abundance at the silent *VSG-6*, active *VSG-2* and procyclin loci, respectively. Red circles, promoters; red boxes, *VSGs*; grey boxes, other genes. RPKM/RPM, reads per (kilobase per) million. **f** Immunofluorescence and protein-blot analysis of pol-I expression and localisation following VEX2 RNAi. The bar graph indicates the proportion of nuclei ($n \geq 100$) with nucleolar and/or extranucleolar pol-I signals; the remaining cells displayed no detectable signal. DNA was counter-stained with DAPI; the images correspond to maximal 3D projections of 0.1 μm stacks; scale bars 2 μm. Error bar, SD; *$p < 0.05$; **$p < 0.01$ (two-tailed paired Student's *t* test); data are averages from three independent biological replicates and representative of independent experiments. Source data are provided as a Source Data file

respectively. Even the *ESAGs* at the active transcribed locus ($n = 18$) were significantly ($p < 10^{-50}$) upregulated (9-fold, Fig. 4d), as were the non-telomeric, pol-I transcribed *procyclin* (associated) genes (Fig. 4e); *procyclin* genes produce abundant surface proteins that are normally expressed only in insect mid-gut stage cells. Highlighting the magnitude of the VEX2-knockdown phenotype, derepressed *VSGs* contributed >24% of total *VSG* transcripts, compared to only 0.1% in control cells (Supplementary Data 1, sheets 4–5); the super-abundant active *VSG* mRNA is >200-fold more abundant than the average mRNA encoding ribosomal proteins. Notably, several previously silent *VSGs* in ESs with dual promoters (*VSG-6, 8, 15* and *17*; see ref. [7]) were particularly derepressed and produced the four most abundant cellular mRNAs, after *VSG-2*; these VSGs were also readily detected at the cell-surface following VEX2-knockdown (Fig. 3e). The same *VSG*-ESs were activated at high rates during VSG switching[27]. Thus, an additional *VSG*-ES promoter likely facilitates the derepression or activation of a *VSG*-ES. We conclude that VEX2 coordinates *VSG* allelic exclusion and has a profound impact on differential *VSG* expression.

We did not anticipate increased expression of *ESAGs* at the active *VSG*-ES in response to VEX2-knockdown (Fig. 4d). However, it has long been known that *ESAGs* upstream of *VSGs*, despite co-transcription in the same polycistron, yield far less abundant transcripts, and no factor responsible for this differential control had been identified previously; *VSG* mRNA

is >140-fold more abundant than the mean *ESAG* mRNA[18]. Our transcriptome analysis now indicates post- or co-transcriptional suppression of these *ESAGs* mediated by VEX2. We speculate, based on sequence similarity of VEX2 to UPF1, that negative control by VEX2 may be related to nonsense-mediated mRNA-decay.

We next asked what impact VEX2-knockdown and *VSG*-ES derepression had on RNA pol-I localisation. Immunofluorescence microscopy revealed a remarkable loss of the nucleolar pol-I signal after only 24 h of knockdown and a substantial loss of detectable extranucleolar pol-I foci; protein blotting indicated that pol-I was still present following VEX2 knockdown (Fig. 4f). These results suggest that derepressed *VSG*-ESs deplete the nucleolus and the ESB of pol-I, redistributing the polymerase to multiple extranucleolar sites.

**VEX-complex knockdown yields a severe exclusion defect.** VEX1 knockdown yielded a moderate *VSG* derepression defect[17] relative to the VEX2 knockdown phenotype (Figs. 3 and 4). We, therefore, knocked down both VEX1 and VEX2 simultaneously and carried out a similar analysis to that described above. VEX1–VEX2 double knockdown was associated with a very severe growth defect that was cytocidal after 72 h (Fig. 5a). In these cells, silent *VSG-6* was again strongly derepressed, as assessed by protein immunoblotting; the active-VSG signal was

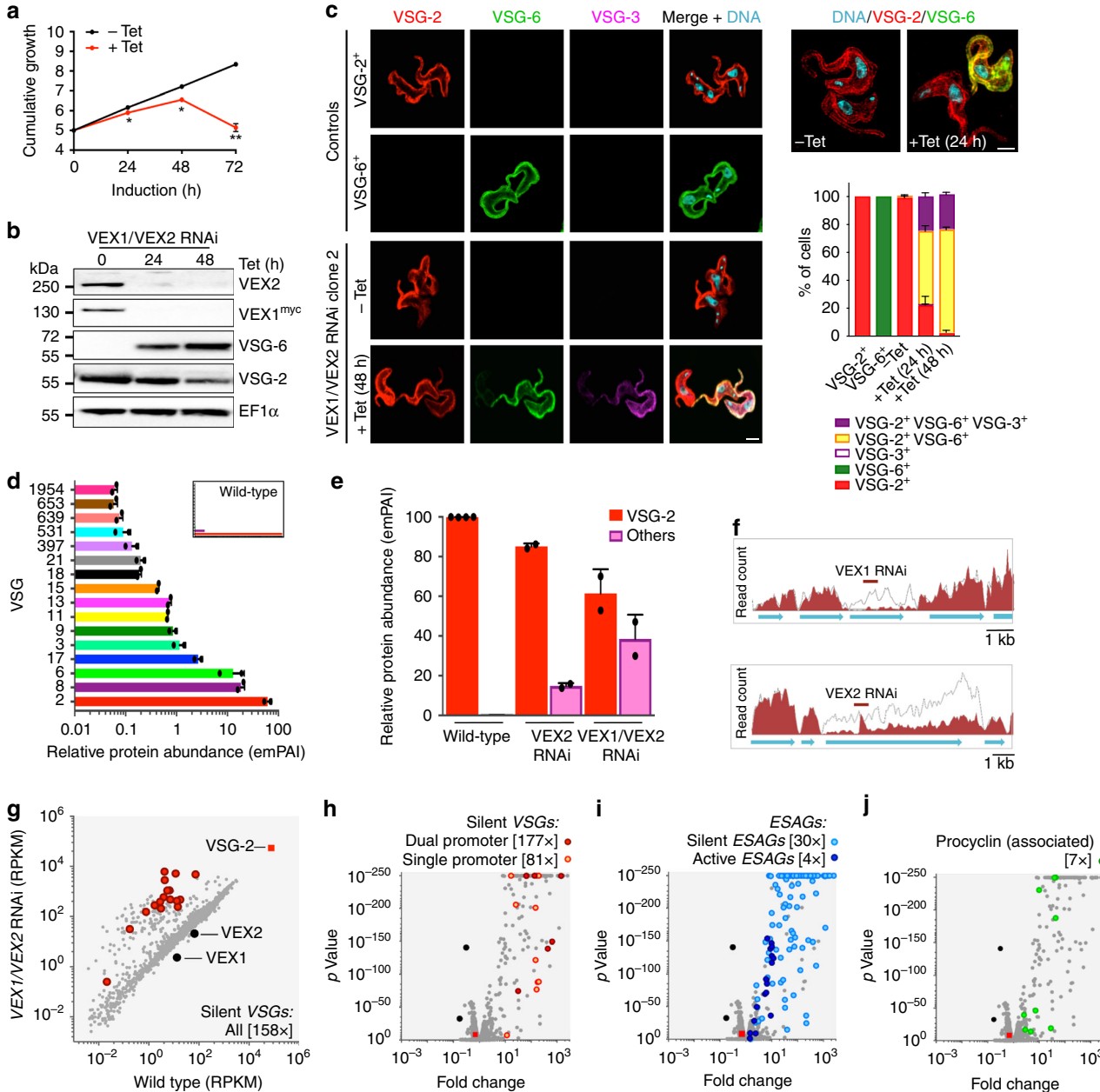

**Fig. 5** VEX-complex knockdown yields a severe exclusion defect. **a** Cumulative growth following tetracycline (Tet) induced VEX1-VEX2 double knockdown; *$p < 0.05$; **$p < 0.01$ (multiple $t$ tests). Protein blotting (**b**), and immunofluorescence analysis (**c**), to assess *VSG* expression following VEX1–VEX2 double knockdown (24 h). Other details as in Fig. 3a–c. In **c**, an additional silent VSG, VSG-3, was analysed. **d** Quantitative mass spectrometry analysis of surface-VSGs following VEX1-VEX2 double knockdown (48 h). The inset shows data for wild-type cells. **e** Relative abundance of surface-VSGs following VEX2 or VEX1/VEX2-knockdown (48 h). emPAI, exponentially modified Protein Abundance Index (**d**, **e**). Data are averages from two independent biological replicates (**a**, **c**–**e**) and representative of independent experiments (**a**–**c**). Error bars, SD. **f**–**j** RNA-seq analysis following VEX1–VEX2 double knockdown (24 h). Total number of reads across *VEX1* and *VEX2* loci in the parental population or following knockdown (**f**). In **g**–**j** specific cohorts of genes are highlighted and values are averages from three independent RNAi strains relative to wild-type controls. Numbers in square brackets indicate increase in total transcript abundance for each category; RPKM, reads per kilobase per million. Source data are provided as a Source Data file

once again clearly reduced after 48 h of knockdown (Fig. 5b). Following 48 h of VEX1–VEX2 double knockdown, immunofluorescence microscopy revealed that almost all cells expressed both VSG-2 and VSG-6, while >20% of cells simultaneously expressed VSG-2, VSG-6 and VSG-3 (Fig. 5c); VSG-13 was also expressed by >20% of these cells (Supplementary Fig. 6a). We further monitored VEX1-knockdown cells over an extended period, during which cells remain viable[17]. This revealed >25% of the population expressing both VSG-2 and VSG-6 throughout the

time-course, with no evidence for switching to VSG-6 expression (Supplementary Fig. 6b). Thus, VEX-complex knockdown disrupts allelic exclusion and does not simply increase the VSG switching rate. Quantitative proteomic analysis following VEX1–VEX2 knockdown revealed derepression of 15 pol-I promoter-associated VSGs (Fig. 5d, Supplementary Data 1, sheet 3). A comparison of relative VSG expression revealed that derepressed VSGs contributed approx. 40% of total VSG following VEX1–VEX2 knockdown (Fig. 5e, Supplementary Data 1, sheet

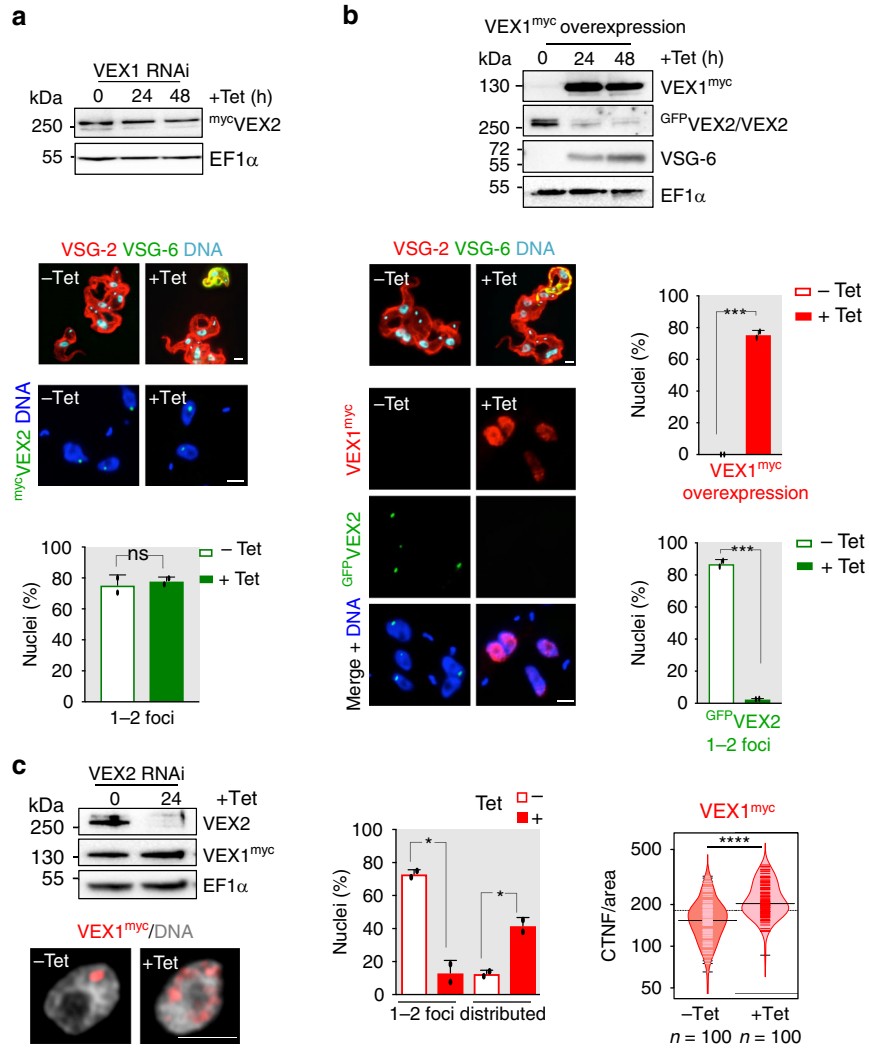

**Fig. 6** VEX-complex interactions impact abundance and location. **a**, **b** Protein-blot and immunofluorescence analysis following Tet-induced VEX1 knockdown and Tet-induced $^{myc}$VEX1 overexpression, respectively (48 h). In the upper immunofluorescence panels, cells were stained with α-VSG-2 and α-VSG-6 to confirm the dual VSG expression phenotypes reported previously[17]. In the lower panels, cells were stained with α-myc for VEX2 localisation (**a**) or both α-myc and α-GFP for VEX1 and VEX2 localisation, respectively (**b**). The bar graphs indicate the proportion of ≥300 nuclei with 1–2 $^{myc}$VEX2 foci following 48 h of VEX1 knockdown (**a**) or with 1–2 $^{GFP}$VEX2 foci following 48 h of VEX1 overexpression (**b**). The proportion of nuclei displaying a distributed VEX1 signal (due to overexpression) is also shown in **b**; the remaining cells displayed no detectable signal. **c** Protein-blot and immunofluorescence analysis of VEX1$^{Myc}$ following Tet-induced VEX2 knockdown (24 h). The bar graph indicates the proportion of nuclei with 1–2 VEX1$^{myc}$ foci or distributed signal, the remaining cells displayed no detectable signal. The bean-plot shows signal intensity in ≥100 nuclei with 1–2 major VEX1$^{myc}$ foci (−Tet) or nuclei with a distributed signal (+Tet 24 h) following VEX2 knockdown. Black lines show the medians; pink/red lines represent individual data points; polygons represent the estimated density of the data. CTNF corrected total nuclear fluorescence, normalised to the nuclear area. EF1α was used as a loading-control. DNA was counter-stained with DAPI; scale bars 2 μm. The images in (**c**) correspond to maximal 3D projections of 0.1 μm stacks. For bar graphs, data representative of independent experiments and averages from two independent biological replicates; error bars, SD. ns, non-significant; *$p < 0.05$; ***$p < 0.001$; ****$p < 0.0001$ (two-tailed paired and unpaired Student's $t$ tests applied to bar graphs and bean plot, respectively). Source data are provided as a Source Data file

3) compared to only approximately 10% of total VSG following VEX2 knockdown. As for VEX2-knockdown (Fig. 4f), double knockdown was associated with loss of both the nucleolar pol-I signal and detectable extranucleolar pol-I foci (Supplementary Fig. 6c).

Transcriptome analysis following VEX1–VEX2 knockdown (Fig. 5f, Supplementary Fig. 7, Supplementary Data 1, sheets 4–5) revealed a similar pattern of derepression as observed above for VEX2 knockdown alone (Fig. 5g–j). Again, all known pol-I transcribed protein-coding loci were derepressed, and previously silent *VSGs* linked to dual *VSG*-ES promoters produced the four most abundant cellular mRNAs, after *VSG-2* (Fig. 5g). These data demonstrate that the VEX-complex coordinates *VSG* allelic exclusion. The allelic exclusion system collapses following depletion of the VEX-complex, yielding multi-*VSG* expression, highly heterogeneous VSG coats and cell-death.

**VEX-complex interactions impact abundance and location.** We next asked what impact VEX1 and VEX2 have on each other. VEX1-knockdown had no detectable impact on VEX2 expression or focal localisation (Fig. 6a). In contrast, VEX1 over-expression substantially diminished VEX2 abundance; VEX1 was highly overexpressed in most nuclei (~80%) and VEX2 foci were no longer detectable by immunofluorescence (Fig. 6b). These data

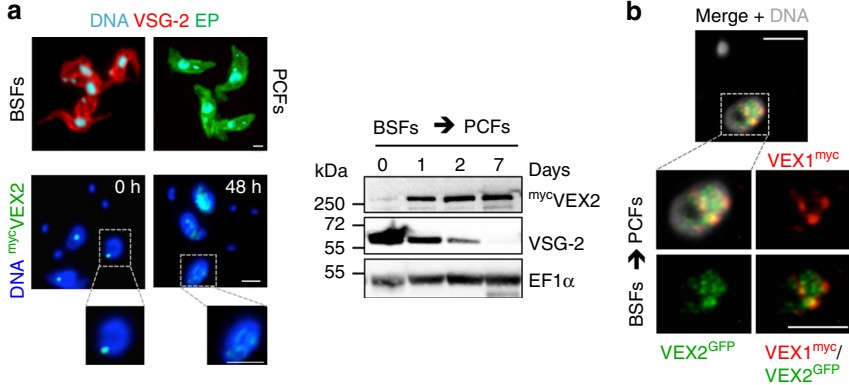

**Fig. 7** VEX-complex compartmentalisation is stage-specific. **a** Immunofluorescence and protein-blot analysis of ^myc^VEX2 localisation and expression, respectively, following differentiation of bloodstream forms (BSFs, 0 h) into procyclic forms (PCFs). As immunofluorescence control, the cells were stained for the major surface glycoprotein for each developmental stage; VSG in BSFs and EP-procyclin in PCFs. For protein blotting, EF1α was used as a loading-control. **b** ^GFP^VEX2/VEX1^myc^ colocalisation studies in 48 h differentiated BSFs. In **b**, these images correspond to maximal 3D projections of 0.1 μm stacks. DNA was counter-stained with DAPI; scale bars 2 μm. Data representative of independent experiments. Source data are provided as a Source Data file

likely explain the previously reported more severe *VSG* derepression phenotype following VEX1 over-expression compared with knockdown[17] and also indicate that VEX1 can limit VEX2 abundance; this latter feature may be important for preventing activation of a second *VSG*. We also tested the impact of VEX2-knockdown on VEX1 localisation and found that VEX1 was redistributed across multiple sub-nuclear puncta following loss of VEX2 (Fig. 6c). Thus, VEX1 association with the *VSG* transcription compartment is VEX2-dependent.

**VEX-complex compartmentalisation is stage-specific**. Insect-stage *T. brucei* cells neither transcribe *VSG* nor exhibit allelic exclusion. Since we show above that VEX-complex association with the *VSG* transcription compartment requires pol-I transcription, we asked whether VEX2 association with this compartment is absent in insect-stage cells. Like VEX1[17], VEX2 was indeed redistributed through the nuclear compartment during differentiation (Fig. 7a). Protein blotting revealed substantial upregulation of VEX2 as the VSG signal diminished (Fig. 7a) and these differentiated cells displayed multiple VEX2-foci, some of which were coincident with VEX1-foci (Fig. 7b). Redistributed VEX2 was similarly observed in long-term established insect-stage cells. Thus, both VEX1 and VEX2 associate with the *VSG* transcription compartment in a pol-I transcription-dependent and life cycle stage-specific manner.

**Colocalisation of CAF-1 and the VEX-complex during S phase**. We next considered the interaction between the VEX-complex and CAF-1; all three components of this conserved chromatin chaperone were co-immunoprecipitated with VEX1 (see Fig. 1b). DNA replication presents both an opportunity to retain or reset epigenetic states. We, therefore, asked if CAF-1 plays a role in the inheritance of the VEX-complex-dependent epigenetic state during DNA replication; the active *VSG*-ES is replicated early during S-phase while the silent *VSG*-ESs are replicated later[28]. Significantly, S-phase specific *VSG*-ES derepression was previously reported following CAF-1b knockdown[15]. By analysing the localisation of a tagged CAF-1b subunit, either CAF-1b^GFP^ or CAF-1b^myc^, we observed major CAF-1b foci, despite an overall punctate nuclear distribution. Notably, these foci co-localised with the sub-nuclear compartment defined by both VEX1 (Fig. 8a, Supplementary Movie 2) and VEX2 (Fig. 8b) and coincident signals were significantly enriched in S-phase cells (Fig. 8a, b); the interaction between the VEX-complex and CAF-1

was also confirmed by immunoprecipitation followed by protein blotting (Fig. 8c).

**VEX-complex reassembly and inheritance requires CAF-1**. By analysing phenotypes associated with CAF-1b knockdown, we confirmed two further specific predictions in support of CAF-1 dependent inheritance of the VEX-complex. First, VEX1 distributed across multiple sub-nuclear puncta following CAF-1b knockdown (12 h, Fig. 9a); while VEX2 notably remained primarily compartmentalised in a single focus in each nucleus under the same conditions (Fig. 9b). Indeed, we observed more intense VEX2 foci following CAF-1b knockdown, and protein blotting confirmed that VEX2 abundance is specifically increased when CAF-1b is depleted (Fig. 9c, Supplementary Fig. 8a, b). Thus, CAF-1 limits VEX2 abundance, an effect that may be enhanced by excess VEX1 (see above). Second, following CAF-1b knockdown (see Fig. 9d), transcriptome analysis revealed specific derepression of *VSGs* and other pol-I transcribed protein-coding genes (Fig. 9e–i, Supplementary Fig. 8c, d, Supplementary Data 1, sheets 4–5).

Following CAF-1b knockdown for 24 h (see Supplementary Fig. 8c), 115 genes displayed >5-fold and significantly ($p < 10^{-2}$) increased expression, >80% of which were also significantly increased following VEX2 knockdown (Supplementary Data 1, sheets 4, 5). In this case, we observed a more pronounced increase in total transcript abundance for promoter-proximal *VSGs* (89-fold, $n = 5$, Fig. 9g) relative to promoter-distal *VSGs* (9-fold, $n = 5$, Fig. 9g). Similarly, we observed a more pronounced increase in total transcript abundance for promoter-proximal *ESAGs* (48-fold, $n = 34$, Fig. 9h) relative to promoter-distal *ESAGs* (4-fold, $n = 96$, Fig. 9h). *Procyclin* (associated) genes were also derepressed (6-fold, Fig. 9i), while *ESAGs* at the active locus displayed significantly ($p < 10^{-4}$) reduced expression (44 ± 19%, $n = 12$; Fig. 9h, Supplementary Data 1, sheets 4–5), consistent with increased VEX2 abundance and enhanced VEX2-mediated negative control at this locus (see above). Thus, CAF-1 knockdown has a major impact on pol-I transcribed protein-coding loci; promoter-proximal genes may be particularly susceptible to derepression due to a genome-wide histone chaperone defect.

**A model for *VSG* allelic exclusion by the VEX-complex**. Finally, combining RNA-seq data, we compared the relative contributions of VEX1, VEX2 and CAF1-b to *VSG* silencing (Fig. 10a) and also calculated relative contributions to an allelic exclusion index; the

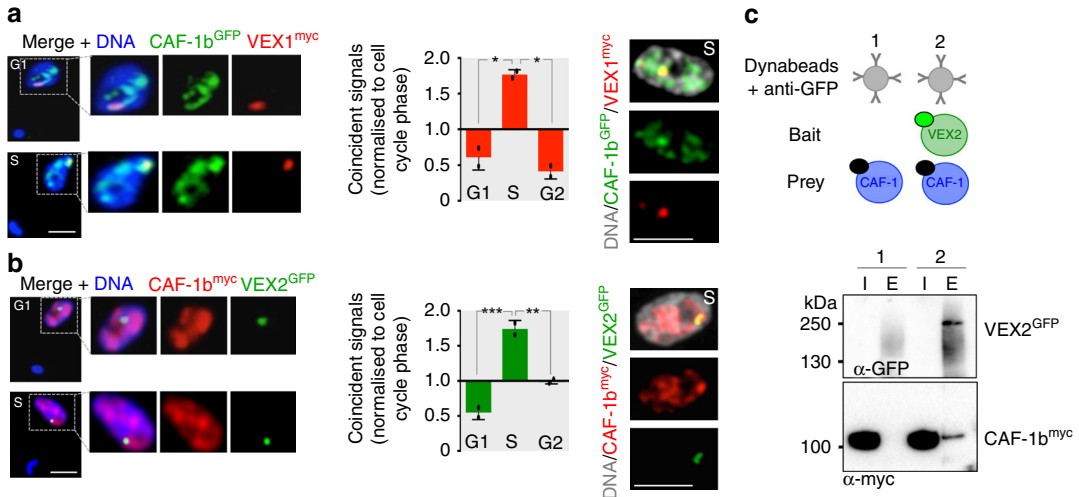

**Fig. 8** Colocalisation of CAF-1 and the VEX-complex during S phase. **a**, **b** Immunofluorescence-based colocalisation studies of VEX1myc/CAF-1bGFP and VEX2GFP/CAF-1bmyc. Representative images from G1 and S phase cells are shown. Nuclei ($n \geq 70$) were scored for CAF-1b puncta coincident with VEX1myc/VEX2GFP foci. DNA was counter-stained with DAPI; scale bars, 2 μm. Error bars, SD; *$p < 0.05$; **$p < 0.01$; ***$p < 0.001$ (one-way ANOVA test). Values are averages from two independent biological replicates and are representative of independent experiments. The images on the right-hand side correspond to maximal 3D projections of 0.1 μm stacks. **c** Co-immunoprecipitation of VEX2GFP/CAF-1bmyc followed by protein-blot analysis. Green circle, GFP; black circles, myc; I, input; E, elution. Source data are provided as a Source Data file

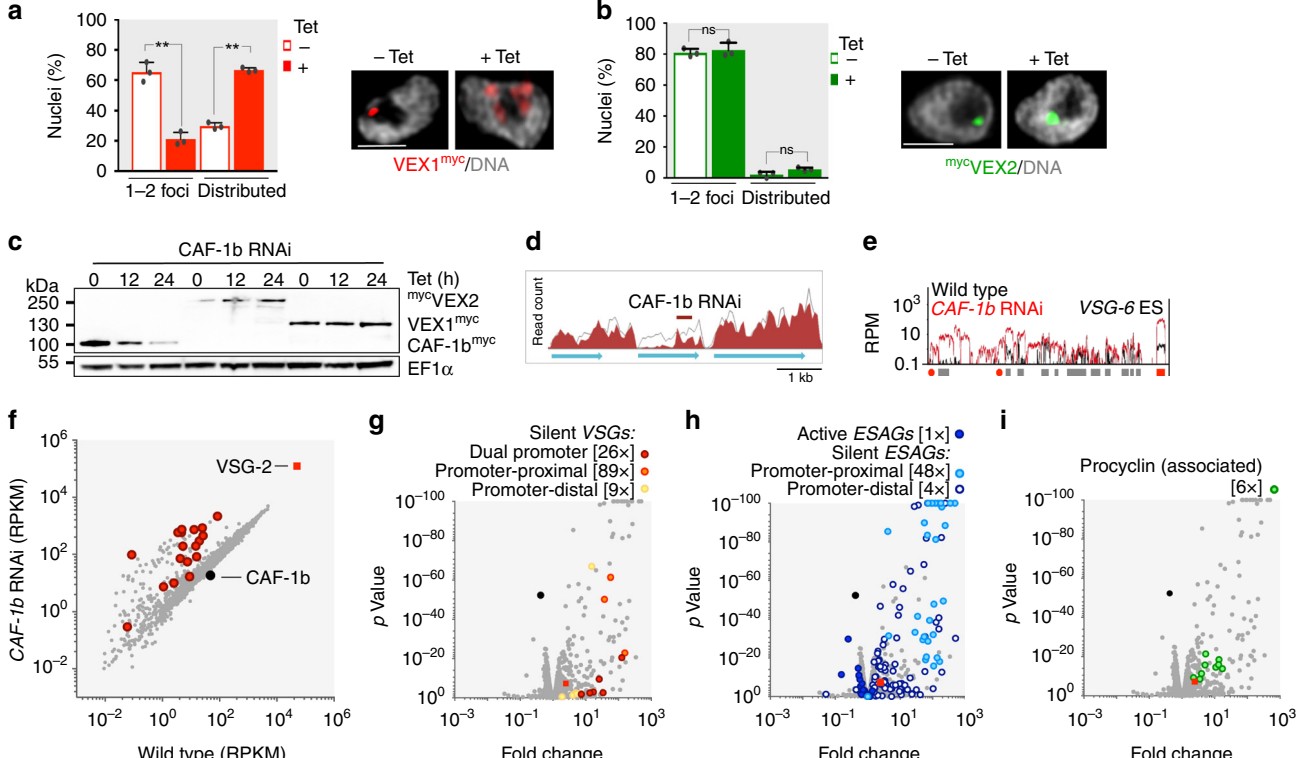

**Fig. 9** VEX-complex reassembly and inheritance requires CAF-1. VEX1myc (**a**), but not mycVEX2 (**b**), is distributed following tetracycline (Tet) induced CAF-1b knockdown (12 h), as assessed by immunofluorescence analysis. DNA was counter-stained with DAPI; scale bars, 2 μm. The images correspond to maximal 3D projections of 0.1 μm stacks. Error bars, SD; ns, not significant; **$p < 0.01$ (two-tailed paired Student's *t* test). Values are averages from three independent biological replicates and are representative of independent experiments (the remaining cells displayed no detectable signal). **c** Protein-blot analysis of CAF-1bmyc, mycVEX2 and VEX1myc expression following CAF-1b knockdown. EF1α was used as a loading-control. **d–i** RNA-seq analysis following CAF-1b knockdown (24 h). **d** Total number of reads across *CAF-1b* locus in the parental population or following *CAF-1b* RNAi. **e–i** Values are averages from two independent RNAi strains relative to wild-type controls. Panel **e** shows transcript abundance at the silent *VSG-6* ES. Red circles, promoters; red boxes, *VSGs*; grey boxes, other genes. RPKM/RPM, reads per (kilobase per) million (**e–i**). Panels **f–i** highlight different gene cohorts and numbers in square brackets indicate increase/decrease in total transcript abundance for each category; red square, *VSG-2*; black circle, *CAF-1b*. Promoter-proximal *VSGs* and *ESAGs* (6, 7 and 10) are within 5 kbp of a promoter, distal *VSGs* are 46 +/−4 kbp from the promoter. Source data are provided as a Source Data file

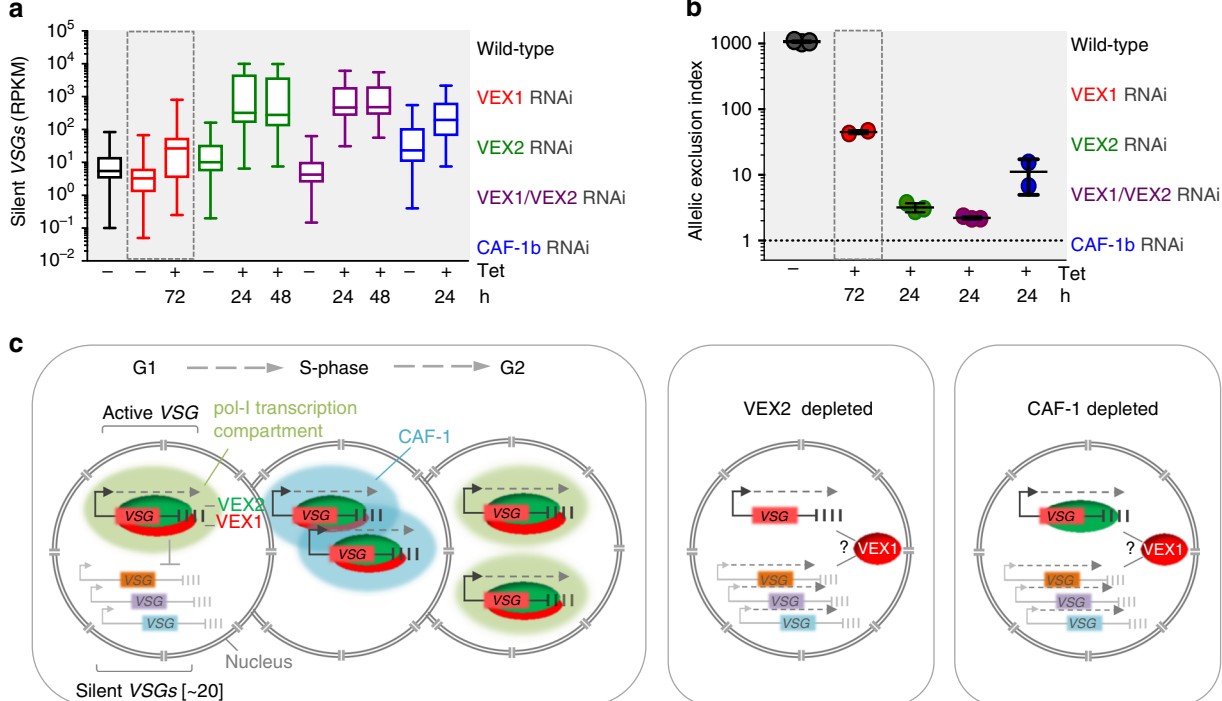

**Fig. 10** A model for *VSG* allelic exclusion by the VEX-complex. **a**, **b** Comparative analysis of *VSG*-transcripts from wild-type *T. brucei* or following VEX1 (data from Glover et al., 2016; dashed boxes), VEX2, VEX1/VEX2 and CAF-1b knockdown (data from current study). **a** The box-plot depicts the sum of silent *VSG* expression; whiskers extend from the minimal to the maximal value, other details as in Fig. 1a. **b** The allelic exclusion index represents active *VSG-2* expression divided by the expression of all other expression site associated *VSGs* combined. Error bars, SD. **c** A model for heritable exclusion by the VEX-complex and CAF-1. Left-hand panel: *VSG*-ES transcription drives competition for a limited pool of the VEX-complex. Once associated with chromatin, the complex promotes transcription and transmits a silencing signal, excluding other *VSG*-ESs, ultimately allowing only one *VSG*-ES to recruit sufficient pol-I and to form a pol-I transcription compartment. CAF-1 dependent compartmentalisation of the complex provides allele-specific epigenetic inheritance. Middle and right-hand panels: VEX2 or CAF-1 depletion disrupts the VEX-complex associated transcription compartment and the silencing signal; it remains unclear how VEX1 interacts with *VSG*-ESs under these circumstances. Bent arrows and dashed arrows indicate *VSG*-ES promoters and transcription, respectively. Source data are provided as a Source Data file

ratio of reads from the active *VSG* relative to all 18 silent expression-site associated *VSGs* (Fig. 10b). VEX2 clearly makes the major contribution to *VSG* silencing (Fig. 10a, b), establishing a 1000-fold expression differential in wild-type cells, which, following VEX2 or VEX-complex depletion, is diminished to a 2–4-fold differential (Fig. 10b). As illustrated by our model (Fig. 10c), we conclude that the VEX-complex associates with the active *VSG* in a pol-I transcription-dependent manner and thereby coordinates nuclear pol-I compartments and *VSG* exclusion. VEX1 compartmentalisation is VEX2-dependent, while retention during DNA replication is also CAF-1 dependent.

## Discussion

An improved understanding of immune evasion by pathogens has profound importance to health and disease. Despite intense study, mechanisms underlying stochastic activation of one allele and the heritable exclusion of all others remain to be fully defined. This has been the case for antigenic variation in the parasites that cause malaria[1], giardiasis[2] and African trypanosomiasis;[3] and also in mammals where olfactory receptor allelic exclusion underpins odour detection[4] and where immunoglobulin and receptor allelic exclusion underpin specificity in B and T cells[5].

Here, we describe an association between a bipartite VSG exclusion (VEX)-complex and the active *VSG* transcription sub-compartment in *T. brucei*. The complex, which is compartmentalised due to active *VSG*-ES transcription, mediates transcriptional silencing of all other *VSG* alleles (Fig. 10c). In

addition, VEX1 sequestration requires VEX2, and also CAF-1 during DNA replication. Transcription-dependent sequestration, and a major allelic exclusion defect when the VEX-complex is depleted are consistent with a sequestration-transcription based positive-feedback mechanism, favoring sequestration of the VEX-complex at the transcribed *VSG* compartment. This may be important for maintaining, and we speculate, also establishing, allelic exclusion. CAF-1 dependence indicates that inheritance of exclusion requires VEX1 reloading during, or soon after, DNA replication. Notably, the VEX-complex is also present in insect-stage cells, suggesting availability for retargeting once *VSG* transcription reinitiates in parasites in the insect salivary gland.

Several observations suggest potentially shared mechanisms with allelic exclusion in mammals. The VEX2-related UPF1 orthologue in humans (Rent1) influences early nuclear events in mRNA biogenesis, including splicing[29]. Mammalian Rent1 is enriched at telomeres and negatively controls the association between telomeric repeat-containing RNA (TERRA) transcripts and chromatin at these sites; it has been proposed that these transcripts promote heterochromatin assembly, similar to *Xist* RNA promotion of X-chromosome inactivation[30]. In addition, murine UPF1 is required for X-inactivation and formation of *Xist* RNA domains, a function involving *Xist* splicing control[31]. Taken together, these findings suggest a conserved role for this helicase family in regulating lncRNA-chromatin associations, and possibly in gene expression choices. The enrichment of VEX1 and VEX2 in association with the active *VSG*-ES is also reminiscent of

nuclear sub-structures in other eukaryotic cells, such as PML bodies and nuclear speckles[32,33].

It remains unclear how the activity state of a gene is reproduced on daughter chromatids following DNA replication. The conserved hetero-trimeric histone H3–H4 specific chaperone, CAF-1, participates in chromatin assembly directly behind the passing DNA polymerase during DNA replication, integrating DNA synthesis with conservative reassembly of chromatin in eukaryotes[24,34]. This involves reincorporating the majority of maternal histones within 400 bp of their pre-replication locus[35] and locus-specific bookmarking by epigenetic regulatory complexes to maintain transcriptional programs[36,37]. Mammalian CAF-1 sustains somatic cell types[38] and epigenetic memory by interacting with heterochromatin protein 1α, a histone H3 methyltransferase and KRAB-ZFP-associated protein 1; this complex is also associated with sub-nuclear foci[39]. Human CAF-1 can also activate transcription independent of its histone deposition function[40] and the conserved CAF-1 subunit-associated dREAM complex plays a role in specifying and maintaining olfactory receptor gene expression in *Drosophila*[41]. Thus, in addition to housekeeping roles in chromatin assembly, several lines of evidence suggest an evolutionary conserved role for CAF-1 in sustaining site-specific molecular memories of epigenetic states. Our findings indicate that this can involve CAF-1 dependent and locus-specific compartmentalisation of non-histone, chromatin-associated regulators as epigenetic marks. Consistent with this view, a cohesin-dependent delay in active *VSG* sister chromatid separation facilitates epigenetic inheritance[42]. Thus, premature segregation of sister chromatids following cohesin depletion may yield free VEX1 and present opportunities for activation of other *VSG*-ESs.

VEX2 makes the greatest contribution to differential VSG expression, as assessed at mRNA and protein levels, while loss of both VEX1 and VEX2 leads to collapse of the exclusion system. Transcription sustains VEX-complex assembly at the active locus and this sequestration through transcription may also drive the establishment of a single active site. The VEX-complex may also be self-limiting, in the sense that CAF-1 and VEX1 can negatively control VEX2 expression; this may be important to minimise activation at other sites and hence secure selectivity. We previously demonstrated homology-dependent silencing by VEX1[17] and suspect that communication among *VSG* alleles requires a *trans*-acting RNA component. Although the current findings demonstrate robust negative control of silent *VSGs* by the VEX-complex, a more detailed dissection of the proposed positive and negative controls, and their coordination to achieve allelic exclusion, will require further study.

In summary, the crosstalk among *VSG*-ESs and inheritance of allelic exclusion in trypanosomes requires transcription, the VEX-complex and CAF-1 dependent partitioning of VEX1. These factors collectively sustain a specific assembly in association with the active *VSG*, resulting in exclusion of all other *VSG* alleles. CAF-1 links the VEX-complex to the active site to produce a molecular memory. Our findings reveal the factors underpinning a winner-takes-all paradigm for the establishment, maintenance and inheritance of nuclear bodies and allele-specific epigenetic states. To our knowledge, this is the first characterisation of a protein complex directly responsible for single gene choice in an allelic exclusion system.

## Methods

**T. brucei growth and manipulation**. Bloodstream-form *T. brucei*, Lister 427 and 2T1 cells[43], both wild-type with respect to VEX1, VEX2 and CAF-1 subunits, were grown in HMI-11 medium and genetically manipulated using electroporation[44]; cytomix was used for all transfections. Puromycin, phleomycin, hygromycin and blasticidin were used at 2, 2, 2.5 and 10 μg ml$^{-1}$ for selection of recombinant

clones; and at 1, 1, 1 and 2 μg ml$^{-1}$ for maintaining those clones, respectively. Cumulative growth curves were generated from cultures seeded at $10^5$ cells ml$^{-1}$, counted on a haemocytometer and diluted back to $10^5$ cells ml$^{-1}$ as necessary. For differentiation of bloodstream form to procyclic form, $2 \times 10^7$ cells were resuspended in DTM medium, 2.5 mg ml$^{-1}$ of haemin, 300 mM cis-aconitate and incubated at 27 °C. Established procyclic-form *T. brucei*, Lister 427 cells were grown in SDM-79 at 27 °C and genetically manipulated using electroporation as above. Blasticidin was used at 10 and 2 μg ml$^{-1}$ for selection and maintenance, respectively.

**Plasmids**. For RNAi, primers were selected from ORF sequences using CLC Viewer v. 7.8 and BLAST analysis to minimise potential off-target effects. A specific RNAi target fragment for VEX2 (Tb927.11.13380, 471 bp) was amplified and cloned into pRPa$^{iSL}$[45]. The VEX1 (Tb927.11.16920, 574 bp)[17] and CAF-1b (Tb927.10.7050, 458 bp)[15] RNAi cassettes were excised prior to electroporation by digesting with *Asc*I. For epitope-tagging at the native locus, a 710 bp fragment of VEX2 was amplified and cloned into pNAT$^{TAGx}$[45] to add an *N*-terminal 6× c-myc or GFP-tag and a fragment of 918 bp was amplified and cloned into pNAT$^{xTAG}$[45] to add a *C*-terminal GFP tag. The vectors were linearised with *Xho*I and *Hpa*I, respectively. The VEX1$^{12myc}$[17] and CAF-1b$^{12myc}$[15] *C*-terminal tagging vectors were linearised with *Sph*I and *Nde*I, respectively. The VEX1 and CAF-1b GFP *C*-terminal tagging vectors were made by replacing the 12× c-myc tag and were also linearised with *Sph*I and *Nde*I, respectively. The VEX1 *C*-terminal myc-tag over-expression cassette[17] was excised prior to electroporation by digesting with *Asc*I. Linearised RNAi and overexpression constructs, under the control of tetracycline-inducible promoters, were transfected into 2T1 cells, which allow for targeting to a single genomic locus validated for robust inducible expression[43].

**ChIP-seq**. ChIP and cell lysis[46] was carried out with the following modifications. Briefly, $2 \times 10^8$ *T. brucei* bloodstream form cells expressing a *C*-terminal 12-myc tagged endogenous copy of *VEX1* were cross-linked with 1% formaldehyde for 20 min at RT. DNA was sonicated using a Bioruptor (Diagenode) with sonication beads (Diagenode, C01020031) for 5 cycles of 30 s on/30 s off. *C*-terminal 12-myc VEX1 was immunoprecipitated with α-Myc antibody coupled to Dynabeads Protein G (2.8 μm). Antibody coupling to the Dynabeads was carried out according to the manufactures recommendations. The beads were then washed with RIPA buffer (50 mM HEPES-KOH, pKa 7.55, 500 mM LiCl, 1 mM EDTA, 1.0% NP-40, 0.7% Na-Deoxycholate) and eluted DNA was purified by phenol chloroform extraction and ethanol precipitation. Reads were mapped to bloodstream *VSG*-ESs[7] and metacyclic *VSG*-ESs[47,48] from the Lister 427 strain. Bowtie 2-mapping[49] was with the parameters --very-sensitive --no-discordant --phred33, and with a MapQ value of >1[18]. Alignment files were manipulated with SAMtools[50]. Alignments were inspected with the Artemis genome browser[51]. Telomeric *VSG* coding sequences with a mapping quality filtering of >1 were aligned using deeptools computeMatrix scale-regions. The bedgraph for log$_2$ fold change was generated using deepTools2[52] with 1 kb bins and the option smoothLength 5000, plotted in Excel and further assembled in Illustrator. The heat map was generated using deepTools. Reads were counted using BEDTools[53], counting reads in non-overlapping 1 kb bins. Files were normalised by counting total read counts per library prior to fold change calculation for each bin. Locus maps were generated by exporting vector graphics views or regions of interest from Artemis genome browser and manipulation in Adobe illustrator. The box plot was generated in Graphpad Prism v7.0.

**Affinity-enrichment of VEX1-interactors**. $1 \times 10^{10}$ insect or bloodstream form *T. brucei* cells with or without a *C*-terminal GFP-tagged endogenous copy of *VEX1* were washed three times in ice cold PBS with protease inhibitors (Roche, EDTA free) and subsequently cryomilled into a fine grindate, in a planetary ball mill (Retsch)[20]; protein–protein interactions are effectively preserved under these conditions. Six aliquots of 50 mg of frozen grindate were thawed into ice cold lysis buffer (20 mM HEPES, pH 7.4, 1 mM MgCl$_2$, 10 μM CaCl$_2$, protease inhibitor cocktail) containing either 100 mM NaCl or 250 mM citrate and either 0.1% Tween, 0.1% Brij58, 0.1% Triton X-100 or 0.1% Chaps, and pipetted to homogenise. The samples were then subjected to sonication (three pulses at 60% amp) and placed on ice. Samples were spun for 10 min at 20,000$g$ at 4 °C to pellet the debris. The supernatant was removed and added to a 1.5 ml eppendorf containing dimeric α-GFP nanobodies[54] conjugated to magnetic beads (Dynabeads—Thermo Scientific) and agitated at 4 °C for 2 h. The samples were then placed on a magnetic rack and washed three times with ice-cold lysis buffer. Samples were eluted with a non-reducing sodium dodecyl sulphate polyacrylamide gel electrophoresis (SDS-PAGE) loading buffer or equivalent. The resulting proteins were fractionated by 1D SDS-PAGE and visualised by silver staining.

**Quantitative mass spectrometry**. Affinity-enriched samples were run 2 cm into a NuPAGE® Novex Bis–Tris 10% gel with NuPAGE® MOPS SDS running buffer (Life Technologies) and subjected to overnight (16 h) trypsin digestion (Modified Sequencing Grade, Roche). Peptides were then extracted, dried in a SpeedVac (Thermo Scientific), resuspended in 50 μl 1% formic acid, centrifuged and transferred to high-performance liquid chromatography (HPLC) vials. 5 μl was typically analysed on a Q Exactive HF Hybrid Quadrupole-Orbitrap Mass Spectrometer

(Thermo Scientific) coupled to an UltiMate 3000 RSLCnano ultra HPLC system (Thermo Scientific) and EasySpray column (75 μm × 50 cm, PepMap RSLC C18 column, 2 μm, 100 Å, Thermo Scientific). The mass spectrometer was operated in data dependent mode with a single MS survey scan from 335 to 1800 $m/z$ followed by 20 sequential $m/z$ dependent MS2 scans. The 20 most intense precursor ions were sequentially fragmented by higher energy collision dissociation. The MS1 isolation window was set to 1.4 Da and the resolution set at 60,000. MS2 resolution was set at 15,000. The AGC targets for MS1 and MS2 were set at $3e^6$ ions and $2e^5$ ions, respectively. The normalised collision energy was set at 27%. The maximum ion injection times for MS1 and MS2 were set at 50 and 19 ms, respectively. The peptides from each fraction were separated using a mix of buffer A (0.1% formic acid in MS grade water) and B (0.08% formic acid in 80% MS grade $CH_3CN$) and eluted from the column using a flow rate of 300 nl min$^{-1}$ and a linear gradient from 5 to 40% buffer B over 68 min. The column temperature was set at 50 °C. RAW data files were extracted and converted to Mascot generic files (.mgf) using MSC Convert. Data were searched against the MaxQuantTbrucei database using the Mascot Search Engine (Mascot Daemon Version 2.3.2). Fold-change, relative to a sample lacking GFP-tagged VEX1, was used to derive an enrichment score. For VSG analysis, glycosylphosphatidylinositol (GPI)-specific phospholipase C (GPI-PLC) cleaved soluble VSG (sVSG) was prepared and subjected to quantitative mass spectrometry analysis[17,55]; the eluate was concentrated on an Amicon Ultra 0.5 ml centrifugal filter (Millipore), and recovered in 100 μl of water. emPAI scores are proportional to protein content in a protein mixture[56].

**Co-Immuniprecipitation**. $4 \times 10^8$ bloodstream form *T. brucei* cells with or without a *C*-terminal GFP-tagged endogenous copy of *VEX2* and a *C*-terminal 12-myc-tagged endogenous copy of *CAF-1b* were washed three times in ice-cold PBS with protease inhibitors and lysed into ice cold lysis buffer (RIPA, 1 mM DTT, protease inhibitor cocktail). Pipetting to homogenise, 30 s of vortexing and incubation for 30 min at 4 °C facilitated lysis. Samples were spun for 10 min at 20,000*g* at 4 °C to pellet debris. The supernatant was removed and added to a 1.5 ml eppendorf containing α-GFP antibody (Abcam) conjugated to magnetic Dynabeads and agitated at 4 °C for 2 h. The samples were then placed on a magnetic rack and washed five times with ice-cold lysis buffer. Samples were eluted with a reducing NuPAGE LDS loading buffer. The resulting proteins were fractionated by 1D SDS-PAGE and analysed by protein-blotting.

**Protein blotting**. Protein samples were run according to standard protein separation procedures, using SDS-PAGE. However, for VEX2 detection, the use of Bis-Tris gels with a neutral pH environment and a bis–tris/bicine based transfer buffer (containing a reducing agent and 10% methanol) were critical for protein separation and transfer, respectively (NuPAGE, Invitrogen). Otherwise, western blotting was carried out according to standard protocols. The following primary antibodies were used: rabbit α-VEX2 (1:1000), rabbit α-pol-I largest subunit[17] (1:500), rabbit α-VSG-2 (1:20,000), rabbit α-VSG-6 (1:20,000), mouse α-c-myc (Millipore, clone 4A6, 1:7,000), rabbit α-GFP (Abcam, 1:1,000) and mouse α-EF1α (Millipore, clone CBP-KK1, 1:20,000). We used horseradish peroxidase coupled secondary antibodies (α-mouse and α-rabbit, Biorad, 1:2000). Blots were developed using an enhanced chemiluminescence kit (Amersham) according to the manufacturer's instructions. Densitometry was performed using Fiji v. 2.0.0.

**Microscopy**. Immunofluorescence microscopy was carried out according to standard protocols, using 12-well 5 mm (Thermo Scientific) or 18-well μ-slides (Ibidi) for wide field, and confocal microscopy, respectively. For super-lowest speed; pinhol microscopy, the cells were attached to poly-L-lysine treated coverslips (#1.5), stained and only then mounted on glass slides. For colocalisation studies with pol-I we used antigen-retrieval. Prior to permeabilization, fixed cells were rehydrated in phosphate-buffered saline (PBS) for 5 min at RT, held at 95 °C for 60 s in freshly prepared antigen-retrieval buffer (100 mM Tris, 5% urea, pH 9.5) and then washed 3 × 5 min in PBS at RT. Cells were mounted in Vectashield with DAPI (wide field) or stained with 1 μg ml$^{-1}$ DAPI for 10 min and then mounted in Vectashield without DAPI (confocal and super resolution). In *T. brucei*, DAPI-stained nuclear and mitochondrial DNA were used as cytological markers for cell-cycle stage; one nucleus and one kinetoplast (1N:1K) indicates G1, one nucleus and an elongated kinetoplast (1N:eK) indicates S-phase, one nucleus and two kineto-plasts (1N:2K) indicates G2/M and two nuclei and two kinetoplasts (2N:2K) indicates post-mitosis[57,58]. Primary antisera were rat α-VSG-2 (1:10,000), rabbit α-VSG-6 (1:10,000), rabbit α-VSG-13[59] (1:5000), mouse α-VSG-3[60] coupled with Alexa 488 (1:500), mouse α-EP procyclin (Cedarlene, 1:1,000), rabbit α-GFP (Invitrogen, 1:250; Abcam, 1:500), mouse α-myc (Source Bioscience, clone 9E10, 1:200 or New England Biolabs, clone 9B11, 1:2,000), rabbit α-pol-I largest subunit[17] (1:100), rabbit α-NOG1[61] (1:500) or rabbit α-histone H3 (Abcam, 1:1000). The secondary antibodies were Alexa Fluor conjugated goat antibodies (Thermo Scientific): α-mouse, α-rat and α-rabbit, AlexaFluor 488 or Alexa Fluor 568 (1:1000 or 1:2000, for confocal or wide field field microscopy, respectively) or α-rat Alexa 647 (1:1000). For the colocalisation studies in Fig. 8a, b and the quantification in Fig. 5c, cells were analysed by confocal microscopy, using a Leica TCS SP8 confocal laser scanning microscope and the Leica Application Suite X (LASX) software

(Leica, Germany). For the remaining quantifications, cells were analysed by wide field microscopy, using a Zeiss Axiovert 200 M microscope with an AxioCam MRm camera and the ZEN Pro software (Carl Zeiss, Germany). The images were acquired as z-stacks (0.1–0.2 μm) and in the case of the wife field microscopy, further deconvolved using the fast iterative algorithm in Zen Pro. For all quantifications, images were acquired with the same settings and equally processed. Corrected total nuclear fluorescence = integrated density − (area of selected nucleus × mean fluorescence of background readings). Representative images were further acquired using a Leica TCS SP8 confocal laser scanning microscope in Hyvolution Mode and the Leica Application Suite X (LASX) software (Leica, Germany). The Hyvolution mode allows super-resolution level images, with settings: highest resolution/lowest speed; pinhole 0.5—Images with DNA in grey (or cyan); Figs. 1d–f, 2a, b, 3c (right hand side) 4f, 5c (right hand side), 6c, 7b, 8a, b (right hand side), 9a, b and Supplementary Fig. 3c, d. All the confocal images (with or without Hyvolution mode) correspond to maximal 3D projections by brightest intensity stacks of approximately 30 slices of 0.1 μm. All the images were processed, scored and the signal quantified Fiji v. 2.0.0[62]. Actinomycin D was applied at 10 μg ml$^{-1}$ for 30 min and pol-I inhibitor (BMH-21) at 1 μM for 30 min, both at 37 °C.

**Flow cytometry**. Flow cytometry was carried out according to standard procedures. The primary antibodies were as follows: rat α-VSG-2 (1:10,000) and rabbit α-VSG-6 (1:10,000). Secondary antibodies were goat α-rat Alexa Fluor 647 and goat α-rabbit Alexa Fluor 488 (both 1:2000). DNA was stained with propidium iodide at 10 μg ml$^{-1}$. Samples were analysed on a BD LSRFortessa (BD Biosciences) and data were visualised using FlowJo software. More than 40,000 events were analysed to determine the percentage of cells in each quadrant.

**Transcriptome analysis**. RNA-seq analysis was performed using 2T1 cells and uninduced or induced clones of VEX2 (24 and 48 h), VEX1/VEX2 (24 and 48 h) or CAF-1b (24 h) RNAi. Briefly, polyadenylated transcripts were enriched using poly-dT beads and reverse-transcribed before sequencing on a HiSeq platform (Illumina). Reads were mapped to a hybrid assembly consisting of the *T. brucei* 927 reference genome plus the bloodstream *VSG*-ESs[7] and metacyclic *VSG*-ESs[47,48] from the Lister 427 strain. Bowtie 2-mapping[49] was with the parameters --very-sensitive --no-discordant --phred33. Alignment files were manipulated with SAMtools[50]. Per-gene read counts were derived using the Artemis genome browser[63], MapQ > 1[18]. Read counts were normalised using edgeR[64] and differential expression was determined with classic edgeR. RPKM values were derived from normalised read counts in edgeR. Base pair resolution plots were generated using an in-house script.

**Statistical analysis**. All statistical analyses were performed using GraphPad Prism Software (version 7.0), except the transcriptomic analysis (described above). Detailed information regarding replicates, statistical tests and outputs can be found in Supplementary Data 1, sheet 6.

**Resources and reagents**. Details of resources and reagents can be found in Supplementary Data 1, sheet 7. All unique materials are available on request.

**Reporting summary**. Further information on research design is available in the Nature Research Reporting Summary linked to this article.

## Data availability
RNA-seq and ChIP-seq data have been deposited in the European Nucleotide Archive www.ebi.ac.uk/ena (Accession nos. PRJEB21615 and PRJEB25352, respectively). Proteomics data have been deposited in the *PRoteomics* IDEntifications (PRIDE) database www.ebi.ac.uk/pride, ProteomeXchange accession no. PXD013304. The source data underlying Figs. 1d, 2a, b, 3a–e, 4f, 5a–e, 6a–c, 7, 8a, b, 9a–c, 10a, b, Supplementary Figs. 3a–f, 4c, 6a–c, 8a, b, d are provided as a Source Data file.

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

## Acknowledgements
We thank T. Owen-Hughes, N. Wiechens and V. Singh for assistance with ChIP-seq library preparation; A. Cassidy for Illumina sequencing; R. Clark of the Flow Cytometry and Cell Sorting Facility at the University of Dundee for cell sorting; D. Lamont, K. Beattie and S. Kosto of the FingerPrints Proteomics Facility at the University of Dundee for assistance with quantitative proteomics; the Dundee Imaging Facility; J. Rouse for access to the Leica Confocal SP8 Hyvolution microscope; S. Alsford (London School of Hygiene & Tropical Medicine) for the CAF-1 RNAi and CAF-1 tagging constructs; Marilyn Parsons (University of Washington) for the α-NOG1 antisera and C. Marques for discussions. The work was funded by Wellcome Trust Investigator Awards to D.H. [100320/Z/12/Z] and M.C.F. [204697/Z/16/Z] with additional support from a Wellcome Trust Centre Award [203134/Z/16/Z]. The University of Dundee Flow Cytometry and Imaging Facilities are supported by a Wellcome Trust award [097418/Z/11/Z], and the MRC Next Generation Optical Microscopy award [MR/K015869/1], respectively; while both the Proteomics and Imaging Facilities are supported by a Wellcome Trust Technology Platform award [097945/B/11/Z].

## Author contributions
Conceptualisation: J.F., L.G., M.C.F. and D.H. Data curation: J.F. and S.H. Investigation: J.F., L.G. and C.B. Supervision: D.H. Original draft: J.F. and D.H. Review and editing: J.F., L.G., S.H., M.C.F. and D.H.

## Additional information

**Competing interests:** The authors declare no competing interests.

