## [Peer Review File · Nature Communications]

Reviewers' Comments:

Reviewer #1:

Remarks to the Author:

How the Variant Surface Glycoprotein multigene family is regulated to achieve monoallelic expression remains one of the holy grails in the Trypanosoma field. In this manuscript, the authors use a combination of biochemical, cellular and genome-wide tools to characterize the role of the complex VEX1-VEX2 and the interacting CAF1 in VSG gene regulation. Pull-downs revealed that these three proteins appear to form a stable complex. Genetic manipulation of the complex components (alone or in combination) revealed that they are all required for monoallelic expression of VSGs.

This manuscript addresses an important question and it is very comprehensive in the analysis of the phenotypes of the VEX1, VEX2 and CAF-1 subunit mutants. The authors clearly show that these three components are very important for VSG gene regulation and their loss from the active VSG, affects transcription from silent VSGs. The mechanism by which a complex sitting in one chromosome locus prevents transcription from other loci remains a difficult question to answer. However, the authors showed nicely that the presence of the VEX1-VEX2 complex at the actively transcribed VSG locus requires active transcription. To gain mechanistic insight, the authors could have tested if VEX1-VEX2 complex interacts with Pol I complex and identify which component makes the "bridge". In the pull-down, did the authors test if VEX1-VEX2 was bound to VSG mRNA, or other potential regulatory lncRNA (as mentioned in the Discussion)? This could also provide some clues of mechanism.

At different moments of the manuscript, the authors take conclusions that are not always supported by their data (see below).

Line 30. "Thus, the VEX-complex coordinates single VSG transcription with VSG-silencing, while CAF-1 chaperone-dependent reloading provides VSG allele-specific epigenetic memory "

The current data does not support this conclusion. Loss of CAF-1, like loss of VEX1 and VEX2, cause monoallelic disruption. Loss of some of these factors cause mislocalization of the others, which is consistent for a complex. However, the authors did not provide data to clearly show how the two phenotypes (mislocalization of the complex and loss of monoallelic expression) come about and how each of the proteins participates in each phenotype. Also, I am not entirely sure what the authors mean by "VSG allele-specific epigenetic memory", but assuming it means keeping a chromatin mark that signals which VSG is actively transcribed, then I would have expected to see experiments that address such chromatin status.

Line 63. "Inheritance of VSG exclusion requires reassembly of the VEX-complex during S-phase, which is mediated by the conserved chromatin assembly factor, CAF-1."

The current data does not support this conclusion. The authors showed that CAF-1 co-localizes in part with VEX1-VEX2 in S-phase, suggesting a role of CAF-1 during DNA replication (which is consistent with CAF-1 role as a chaperone). But the authors did not show, for example, that CAF1 is required to recruit the VEX1-VEX2 complex. The data shows that, once CAF1 is depleted, the complex is no longer maintained at the VSG active site. Recruitment is different from maintenance.

Line 142. "Taken together, these results indicate major disruption of allelic exclusion and the simultaneous expression of multiple VSGs on the surface of individual cells following VEX2-knockdown. "

The authors should be more precise in what they mean by "major disruption allelic exclusion". More importantly, the analysis performed at the single cell level were IFA and FACS (Figure 2C-D) and in both the authors tested the expression of the same two VSGs (VSG2 and VSG6). The transcriptome and proteome analysis were performed at the population level and thus it is possible that, for

example, there are never three or more VSGs at the surface of a single parasite. Could the authors show loss of allelic expression with other VSGs (VSG2 and VSG13, for example)? Could the authors show that three VSGs can be detected at the surface of the same parasite?

Line 189. "Thus, allelic exclusion appears to fail completely following depletion of the VEX complex, yielding multi-VSG expression, highly heterogeneous VSG coats and cell-death. "

The authors cannot conclude that VSG coats are highly heterogeneous because, once again, at the single-cell level only two VSGs were shown to be present at the surface of parasites. - see point above.

Minor points:

Figure 1 - Proteomics was performed in BSF and PCF. It seems that one experiment was performed in BSF and three experiments in PCF. Is this correct? Given that PCF do not express VSG (as explained by authors in lines 124-126), it is somewhat unexpected that the authors were able to purify the same complex (VEX1-VEX2 + CAF1) in the two stages of the life cycle. This should be discussed. In Figure 1, the authors confirm co-localization by IFA, but only in BSF. What were the results of immunolocalization VEX1, VEX2 and CAF-1 in PCF? Figure 4D shows IFA only for VEX2 in PCF.

Figure 2D/2E - FACS analysis shows that 92.5% of cells express VSG2 and VSG6 at the surface. However, quantifications performed by Mass Spectrometry show that the levels of surface VSG6 to be around 10-fold lower than surface VSG2. Can the authors conclude that in cells that express both VSGs, there is one VSG that is much more abundant than the other? If so, this should be clearly stated. As it is, the reader could be misled to think that VSG6 is as abundant as VSG2, especially because the green and red signals of Panel 2C have equal intensities. I understand antibodies are different and intensities cannot be compared, but these technical details are currently not explained and they could be misinterpreted.

Reviewer #2:

Remarks to the Author:

1. Major conclusions of this study are based on the identification of VEX1GFP-associated proteins using quantitative proteomic analysis. Therefore, it would be good to upload the raw data of replicates to assess the quality of proteomics data. From supplementary data it is not clear that identified proteins are based on what data quality (e.g. FDR, Number of peptides etc.).

2. Authors have assembled three inducible RNAi knockdown strains with an active VSG-2 expression site. It would be interesting to perform a quantitative proteomics-based investigation on these strains to further understand the function of VEX2. Further, transcriptome and proteome correlation or integrated data analysis may provide interesting clues.

3. VEX and CAF-1 interaction could be well characterized using Surface Plasmon Resonance.

Overall this study is very interesting but few additional experiments are required to make it conclusive. Further, authors should upload the raw MS data.

Reviewer #3:

Remarks to the Author:

Monoallelic gene expression is a common strategy used in various biological systems, such as immune and olfactory systems as well as immune evasion by parasites, including VSG allelic exclusion in *Trypanosoma brucei*. Glover et al (PNAS 2016) previously identified VEX1, an important protein for monoallelic VSG expression in *T. brucei*. VEX1 protein forms a subnuclear compartment which is very closely located to the ES body (extranucleolar pol I site) where the active VSG is transcribed. Depletion of VEX1 derepressed silent VSGs, and interestingly, VEX1 overexpression also disrupted VSG silencing. Although VEX1 is important for monoallelic gene expression, mechanism was not clear.

To follow up and gain mechanistic understanding, in this study, authors identified VEX1 interacting proteins, (i) an orthologue of UPF1 (nonsense-mediated decay factor), which they named VEX2, and (ii) three subunits of CAF-1 complex, a histone chaperone. VEX2 colocalizes with RNA pol I at ES body. VEX1 and 2 showed partial colocalization. VEX2 KD causes nuclear redistribution of VEX1 (speckled signal of VEX1 by IF), indicating a VEX2-dependent VEX1 compartmentalization. VEX2 KD showed strong VSG derepression and derepression was even stronger when both VEX1/2 proteins were depleted. VEX1 and 2 protein levels were co-regulated so the ratio seems to be important. Additionally, to ensure the same VSG2 is active in daughter cells (as trypanosomes maintain in-situ switching at a very low rate), CAF-1 acts in reloading of VEX1 onto newly replicated sister chromatids. Loss of these factors leads to varying degree of VSG derepression phenotypes, with VEX1/2 double KD exhibiting the strongest phenotype. Taken together, the study concludes that "the VEX complex coordinates single VSG transcription with VSG silencing, while CAF-1 chaperone-dependent reloading provide VSG allele-specific epigenetic memory"

Overall, VSG derepression phenotypes are very interesting and novel in the field. These findings are generally robust and will be valuable in solving puzzles of antigenic variation mechanism in *T. brucei*. Results are pretty complicated, as expected given the complexity of allelic exclusion, and need better explanation in some parts. Conclusion seems a bit vague and not clearly stated, mainly because there are many assumptions. I feel that a couple of additional experiments will help gain clearer molecular understanding – (i) pol I IF in KD cells and (ii) VEX1/2 interaction with ES. Specific concerns are listed below.

Major concerns:

1. Identity of subnuclear compartments: There are at least three compartments; VEX1 alone, VEX1-VEX2 together, and VEX2-pol I together (because VEX1 does not really overlap with ES body. VEX1 pulled down VEX2 but they do not 100% overlap in IF. VEX2 overlaps with ES body). These are not clearly distinguished. For example, it is unclear what is meant by 'a single active VSG-associated subnuclear..'. I would think this is VEX2-pol I, but it might be VEX1-VEX2. It appears that one of reasons why it is not very clear is because there is no pol I information in KD cells, or in cells treated with transcription inhibitors. There is an assumption that VEX2 signal is the same as pol I signal in some case (when VEX2 signal does not change) and it is not the same in other case (when VEX2 signal dispersed). To get clarity, pol I IF needs to be done in KD cells as well.

2. Conclusion on VEX complex's role is not clearly stated: It is unclear whether roles of VEX1/2 complex is to promote transcription of active VSG or to inhibit expression of silent VSGs, or simply acting as a guard. VEX2 localization does not change in the absence of VEX1 and VEX1 distributes all over in the absence of VEX2. But outcomes are similar – silent VSGs are transcribed. How does this happen? There are several explanations. First, VEX1 and VEX2 are guarding the ESB so no silent ES can access to the pol I – therefore promoting transcription of active ES only. I would expect to see one ESB in KD cells in this case. Alternative is that VEX1 and VEX2 can drive transcription but are sequestered in the ESB, only for the active VSG. In VEX2 KD cells, VEX1 forms multiple foci, where silent ESs might be transcribed. Glover et al showed that VEX1 forms two foci when two VSGs are

active in an artificial system. It is possible that redistributed multiple VEX1 loci may drive transcription of silent VSGs. VEX2 colocalizes with pol I, so it is possible that integrity of ES body may be disrupted in VEX2 KD cells, in which case, number of pol I foci may increase in the absence of VEX2. To get clarify, I suggest an experiment, in addition to pol I localization in KD cells as described above, VEX proteins-ES interaction in wild type and KD cells (e.g., ChIP and q-PCR).

3. It is amazing that these KD cells do not switch. This implies that the ESB is still occupied by the VSG2 ES, assuming that there is only one ESB pol I. If a silent BES occupies the ESB pol I, this should induce in situ switching, therefore VSG2 protein level should be reduced greatly, which was not the case from Figure 2d. However, Pinger et al (2017 Nat Comm) showed that after switching, old VSG half-life is about 5 hours with regular growth rate taken into account. As VEX2 KD and double KD cells are not dividing well, it is possible that VSG2 half-life could be even longer in switched cells. So, although these cells have ESB occupied by a silent ES(s), VSG2 is a major surface protein. I wonder whether relocalization of VEX proteins is a transition step during in-situ switching. If KD cells have cell cycle arrest, they may not complete the switching process (?). Do any of KD cells have cell cycle defects? Have you examined longer time points? If so, any switchers observed?

4. VSG switching seems to be semi-hierarchical. Is there any preference for VSG6 or any of those highly derepressed VSGs during switching? Have you done IF with other VSGs, such as VSG-8 or VSG-17?

5. It is very interesting that VEX proteins redistribute during transition from BF to PF. Therefore, redistributed VEX proteins may represent a snapshot of normal in-situ switching process. This data also indicates a pol I transcription specific VEX localization, which is more convincing than BMH-21 or Act D treatment, as these drugs cause many other effects that are not related to transcription elongation.

6. Protein-protein interaction was shown by IF, which is good because it shows cellular localization. But IP western is more sensitive to study/confirm CAF1-VEX1 interaction, especially because CAF-1 signal is all over the nucleus. Information of pol I-VEX2 interaction in wild type and VEX1 KD cells would be informative as well.

7. It is interesting that highly derepressed VSGs are different in CAF-1 KD cells (strong derepression of metacyclic VSGs), because both CAF-1 and VEX2 RNAi induce VEX1 redistribution. This needs some explanation. Metacyclic VSGs are more accessible to the ESB or redistributed VEX1 prefers metacyclic VSGs in CAF-1 KD? Or something else. Does CAF-1 affect switching rate? CAF-1 depletion may have differential effects on chromatin structure of ES or metacyclic VSG promoters.

8. Epigenetic memory and epigenetic inheritance are not interchangeable. VEX1 reloading by CAF-1 is epigenetic inheritance, not memory. Epigenetic memory would be a situation in which, when VEX KD cells are re-expressing VEX proteins, how these cells would remember and re-activate the original VSG2, instead of choosing random VSGs.

Minor points

Line 32: There are several canonical UPF homologues in *T. brucei* but their roles are not clear. Since the study does not really show any UPF1 related findings, UPF1 should not be emphasized too much.

In Figure 2 d FACS plot, why is VSG2 signal very heterogeneous, ranging from 10²-10⁵? Have you done similar FACS experiments for double KD and CAF-1? Was there any change with VSG2 signal

(e.g., reduction?)

Line 133: "VSG-6 appeared to accumulate in intracellular compartments as well as the surface" But VSG proteomics was done with proteins released from the cell-surface, this may underrepresent silent VSG levels and types.

Line 169: "we did not anticipate derepression of ESAGs at the active VSG" they were not repressed (they are in the active ES), so it should "increased level" or something.

Line 198, 317: "completely fail, complete failure" it is not clear what is "complete" failure. One may think that "complete" failure of allelic exclusion would be expression of silent VSGs in all cells "without having any major VSG". In KD cells, VSG2 still seems to be the major surface coat. FACS plot would be nice to visualize VSG2 signal loss in VEX double KD cells.

Line 221, 282: 'positive feedback' it is not clear what this means. What exactly is feeding back to what?

Figure 4D: this is a better assay to show transcription dependent VEX2 association with ESB, as during BF to PF transition, transcription of the active ES is lost, without compromising other things, unlike drug treatment. Dispersed VEX may capture the next one to be activated, in this case procyclin promoter. You may want to put Figure 1f-1g in supplementary or together with Figure 4.

Line 234-236: To confirm the interaction between CAF-1 and VEX proteins, one can simply do IP western. Tagged proteins and antibodies are available, so this should be simple.

Line 240: although VEX1 redistributed in CAF-1 KD, increase in VEX2 is not due to VEX1 redistribution, because VEX1 RNAi does not reduced VEX2 level. VEX2 level seems to be controlled independently by CAF-1 and VEX1.

Line 283: "VEX-complex... likely important for both establishing and maintaining allelic exclusion" data here is supporting maintenance of exclusion, but not how it is "established".

Line 313: "indeed, a cohesion-dependent delay in active VSG sister chromatid separation facilitates epigenetic inheritance, entirely consistent with a VEX complex reloading step" Derepression of VSGs in cohesion KD was due to the increased rate of in-situ switching, therefore it is not relevant to VEX work.

Reviewer 1:

How the Variant Surface Glycoprotein multigene family is regulated to achieve monoallelic expression remains one of the holy grails in the Trypanosoma field. In this manuscript, the authors use a combination of biochemical, cellular and genome-wide tools to characterize the role of the complex VEX1-VEX2 and the interacting CAF1 in VSG gene regulation. Pull-downs revealed that these three proteins appear to form a stable complex. Genetic manipulation of the complex components (alone or in combination) revealed that they are all required for monoallelic expression of VSGs.

This manuscript addresses an important question and it is very comprehensive in the analysis of the phenotypes of the VEX1, VEX2 and CAF-1 subunit mutants. The authors clearly show that these three components are very important for VSG gene regulation and their loss from the active VSG, affects transcription from silent VSGs.

The mechanism by which a complex sitting in one chromosome locus prevents transcription from other loci remains a difficult question to answer. However, the authors showed nicely that the presence of the VEX1-VEX2 complex at the actively transcribed VSG locus requires active transcription.

1.1: To gain mechanistic insight, the authors could have tested if VEX1-VEX2 complex interacts with Pol I complex and identify which component makes the "bridge".

R1.1: We see no evidence for a direct interaction between the VEX-complex and pol-I. We searched for all 14 pol-I and class I transcription factor A (CITFA) components in the VEX1-interacting proteome data and only found peptides for Tb927.11.18680 - dynein light chain LC8 of CITFA (Kirkham et al., 2015; PMID:26459761). For this protein, we detected 1 peptide match in the control and only 2 peptide matches in the BSF-IP sample (proteins plotted in Fig. 1b and S2a have a min. of 5 peptide matches in the IP). We've added "Notably, no pol-I components were enriched".

1.2: In the pull-down, did the authors test if VEX1-VEX2 was bound to VSG mRNA, or other potential regulatory lncRNA (as mentioned in the Discussion)? This could also provide some clues of mechanism.

R1.2: The pull-down was used only to identify interacting proteins. A different Cross-Linking ImmunoPrecipitation-seq (CLIP-seq) or similar protocol would likely be more suitable to identify interacting RNA. One concern with such an experiment is that we may see non-specific interactions with the active VSG mRNA, for example, simply due to proximity to the active site. As stated in the Discussion, our view is that "A more detailed dissection of [a suspected *trans*-acting RNA component] will require further study".

At different moments of the manuscript, the authors take conclusions that are not always supported by their data (see below).

1.3: Line 30. "Thus, the VEX-complex coordinates single VSG transcription with VSG-silencing, while CAF-1 chaperone-dependent reloading provides VSG allele-specific epigenetic memory". The current data does not support this conclusion. Loss of CAF-1, like loss of VEX1 and VEX2, cause monoallelic disruption. Loss of some of these factors cause mislocalization of the others, which is consistent for a complex. However, the authors did not provide data to clearly show how the two phenotypes (mislocalization of the complex and loss of monoallelic expression) come about and how each of the proteins participates in each phenotype. Also, I am not entirely sure what the authors mean by "VSG allele-specific epigenetic memory", but assuming it means keeping a chromatin mark that signals which VSG is actively transcribed, then I would have expected to see experiments that address such chromatin status.

R1.3: We've edited this text to more accurately reflect our findings. We replaced "memory" with "inheritance" in the title and abstract. We have also now added ChIP-seq data for VEX1 and all VSG-ESs, revealing enrichment at the active VSG (Fig. 1a, Supplementary Fig. 1, Supplementary Data File S1, sheet 1). Thus, VEX1 does indeed appear to be a chromatin mark that signals which VSG is actively transcribed.

1.4: Line 63. "Inheritance of VSG exclusion requires reassembly of the VEX-complex during S-phase, which is mediated by the conserved chromatin assembly factor, CAF-1." The current data does not support this conclusion. The authors showed that CAF-1 co-localizes in part with VEX1-VEX2 in S-phase, suggesting a role of CAF-1 during DNA replication (which is consistent with CAF-1 role as a chaperone). But the authors did not show, for example, that CAF1 is required to recruit the VEX1-VEX2 complex. The data shows that, once CAF1 is depleted, the complex is no longer maintained at the VSG active site. Recruitment is different from maintenance.

R1.4: We've edited this text at the end of the Introduction to more accurately reflect our findings.

1.5: Line 142. "Taken together, these results indicate major disruption of allelic exclusion and the simultaneous expression of multiple VSGs on the surface of individual cells following VEX2-knockdown." The authors should be more precise in what they mean by "major disruption allelic exclusion". More importantly, the analysis performed at the single cell level were IFA and FACS (Figure 2C-D) and in both the authors tested the expression of the same two VSGs (VSG2 and VSG6). The transcriptome and proteome analysis were performed at the population level and thus it is possible that, for example, there are never three or more VSGs at the surface of a single parasite. Could the authors show loss of allelic expression with other VSGs (VSG2 and VSG13, for example)? Could the authors show that three VSGs can be detected at the surface of the same parasite?

R1.5: We now show loss of allelic exclusion by IFA using two additional VSGs, VSG-3 (Fig. 3c) and VSG-13 (Fig. S6a). We could indeed show that three VSGs can be detected at the surface of the same parasite following VEX-complex knockdown (Fig. 3c).

1.6: Line 189. "Thus, allelic exclusion appears to fail completely following depletion of the VEX complex, yielding multi-VSG expression, highly heterogeneous VSG coats and cell-death." The authors cannot conclude that VSG coats are highly heterogeneous because, once again, at the single-cell level only two VSGs were shown to be present at the surface of parasites. - see point above.

R1.6: See response R1.5 above. We have replaced "fail completely" with "collapse(s)".

Minor points:

1.7: Figure 1 - Proteomics was performed in BSF and PCF. It seems that one experiment was performed in BSF and three experiments in PCF. Is this correct? Given that PCF do not express VSG (as explained by authors in lines 124-126), it is somewhat unexpected that the authors were able to purify the same complex (VEX1-VEX2 + CAF1) in the two stages of the life cycle. This should be discussed. In Figure 1, the authors confirm co-localization by IFA, but only in BSF. What were the results of immunolocalization VEX1, VEX2 and CAF-1 in PCF? Figure 4D shows IFA only for VEX2 in PCF.

R1.7: This is correct. We've now added, "Notably, the VEX-complex is also present in insect-stage cells, suggesting availability for retargeting once VSG transcription reinitiates in parasites in the insect salivary gland". We've also now added dual VEX1-VEX2 IFA data for PCF (insect-stage) cells (Fig. 4d, inset).

1.8: Figure 2D/2E - FACS analysis shows that 92.5% of cells express VSG2 and VSG6 at the surface. However, quantifications performed by Mass Spectrometry show that the levels of surface VSG6 to be around 10-fold lower than surface VSG2. Can the authors conclude that in cells that express both VSGs, there is one VSG that is much more abundant than the other? If so, this should be clearly stated. As it is, the reader could be misled to think that VSG6 is as abundant as VSG2, especially because the green and red signals of Panel 2C have equal intensities. I understand antibodies are different and intensities cannot be compared, but these technical details are currently not explained and they could be misinterpreted.

R1.8: We've added, "VSG-2 remained the most abundant".

Reviewer 2:

2.1: Major conclusions of this study are based on the identification of VEX1GFP-associated proteins using quantitative proteomic analysis. Therefore, it would be

good to upload the raw data of replicates to assess the quality of proteomics data. From supplementary data it is not clear that identified proteins are based on what data quality (e.g. FDR, Number of peptides etc.).

R2.1: We do show 'score', 'num. of (significant) matches', 'num. of (significant) sequences', 'emPAI' and 'enrichment' scores for all four datasets (Supplementary Data File S1, sheet 2). The raw data have been deposited in the PRIDE database.

2.2: Authors have assembled three inducible RNAi knockdown strains with an active VSG-2 expression site. It would be interesting to perform a quantitative proteomics-based investigation on these strains to further understand the function of VEX2. Further, transcriptome and proteome correlation or integrated data analysis may provide interesting clues.

R2.2: For all knockdown strains, we sampled global expression changes using transcriptomics. These data revealed highly specific derepression of pol-I transcription units in every case. The findings did not suggest any direct impact of VEX2 on translation, for example. It is also important to note that global proteomic analyses may fail to detect low-abundance proteins, such as VEX1 (see Urbaniak *et al.*, 2012; PMID: 22574199, for example). We, therefore, focused on surface VSGs for our proteomics analyses, which allowed us to optimize the quantitative dynamic range for these key proteins.

2.3: VEX and CAF-1 interaction could be well characterized using Surface Plasmon Resonance.

R2.3: We have initiated work to express recombinant VEX proteins and CAF-1 subunits. Unfortunately, none of the CAF-1 subunits is currently available in recombinant form. As an alternative to SPR, we have added data confirming interaction between the VEX-complex and CAF-1, using IP and protein blotting (Fig. 5c; also see 3.6 and 3.16 below).

Overall this study is very interesting but few additional experiments are required to make it conclusive. Further, authors should upload the raw MS data.

Reviewer 3:

Monoallelic gene expression is a common strategy used in various biological systems, such as immune and olfactory systems as well as immune evasion by parasites, including VSG allelic exclusion in *Trypanosoma brucei*. Glover *et al* (PNAS 2016) previously identified VEX1, an important protein for monoallelic VSG expression in *T. brucei*. VEX1 protein forms a subnuclear compartment which is very closely located to the ES body (extranucleolar pol I site) where the active VSG is transcribed. Depletion of VEX1 derepressed silent VSGs, and interestingly, VEX1 overexpression also disrupted VSG silencing. Although VEX1 is important for monoallelic gene expression, mechanism was not clear.

To follow up and gain mechanistic understanding, in this study, authors identified VEX1 interacting proteins, (i) an orthologue of UPF1 (nonsense-mediated decay factor), which they named VEX2, and (ii) three subunits of CAF-1 complex, a histone chaperone. VEX2 colocalizes with RNA pol I at ES body. VEX1 and 2 showed partial colocalization. VEX2 KD causes nuclear redistribution of VEX1 (speckled signal of VEX1 by IF), indicating a VEX2-dependent VEX1 compartmentalization. VEX2 KD showed strong VSG derepression and derepression was even stronger when both VEX1/2 proteins were depleted. VEX1 and 2 protein levels were co-regulated so the ratio seems to be important. Additionally, to ensure the same VSG2 is active in daughter cells (as trypanosomes maintain in-situ switching at a very low rate), CAF-1 acts in reloading of VEX1 onto newly replicated sister chromatids. Loss of these factors leads to varying degree of VSG derepression phenotypes, with VEX1/2

double KD exhibiting the strongest phenotype. Taken together, the study concludes that “the VEX complex coordinates single VSG transcription with VSG silencing, while CAF-1 chaperone-dependent reloading provide VSG allele-specific epigenetic memory”

Overall, VSG derepression phenotypes are very interesting and novel in the field. These findings are generally robust and will be valuable in solving puzzles of antigenic variation mechanism in *T. brucei*. Results are pretty complicated, as expected given the complexity of allelic exclusion, and need better explanation in some parts. Conclusion seems a bit vague and not clearly stated, mainly because there are many assumptions. I feel that a couple of additional experiments will help gain clearer molecular understanding – (i) pol I IF in KD cells and (ii) VEX1/2 interaction with ES. Specific concerns are listed below.

(i) See 3.1 below. (ii) See 3.2 below. We've also added a 'model' (Fig. 6c) to provide a better explanation of our results.

Major concerns:

3.1: Identity of subnuclear compartments: There are at least three compartments; VEX1 alone, VEX1-VEX2 together, and VEX2-pol I together (because VEX1 does not really overlap with ES body. VEX1 pulled down VEX2 but they do not 100% overlap in IF. VEX2 overlaps with ES body). These are not clearly distinguished. For example, it is unclear what is meant by ‘a single active VSG-associated subnuclear..’. I would think this is VEX2-pol I, but it might be VEX1-VEX2. It appears that one of reasons why it is not very clear is because there is no pol I information in KD cells, or in cells treated with transcription inhibitors. There is an assumption that VEX2 signal is the same as pol I signal in some case (when VEX2 signal does not change) and it is not the same in other case (when VEX2 signal dispersed). To get clarity, pol I IF needs to be done in KD cells as well.

R3.1: We agree with this view. i.e. VEX1 appears primarily telomere-associated while VEX2 may link the VEX1 and pol-I compartments. We've adjusted this text in the abstract to clarify. RNA pol-I IFA following VEX2 RNAi has now been done, revealing loss of the nucleolar and extranucleolar foci (Fig. 2k, Supplementary Fig. 6c).

3.2: Conclusion on VEX complex's role is not clearly stated: It is unclear whether roles of VEX1/2 complex is to promote transcription of active VSG or to inhibit expression of silent VSGs, or simply acting as a guard. VEX2 localization does not change in the absence of VEX1 and VEX1 distributes all over in the absence of VEX2. But outcomes are similar – silent VSGs are transcribed. How does this happen? There are several explanations. First, VEX1 and VEX2 are guarding the ESB so no silent ES can access to the pol I – therefore promoting transcription of active ES only. I would expect to see one ESB in KD cells in this case. Alternative is that VEX1 and VEX2 can drive transcription but are sequestered in the ESB, only for the active VSG. In VEX2 KD cells, VEX1 forms multiple foci, where silent ESs might be transcribed. Glover et al showed that VEX1 forms two foci when two VSGs are active in an artificial system. It is possible that redistributed multiple VEX1 loci may drive transcription of silent VSGs. VEX2 colocalizes with pol I, so it is possible that integrity of ES body may be disrupted in VEX2 KD cells, in which case, number of pol I foci may increase in the absence of VEX2. To get clarify, I suggest an experiment, in addition to pol I localization in KD cells as described above, VEX proteins-ES interaction in wild type and KD cells (e.g., ChIP and q-PCR).

R3.2: The integrity of the ESB is indeed disrupted after VEX2 knockdown (Fig. 2k, Supplementary Fig. 6c). While the data do not support a 'guarding' model, a scenario whereby redistributed VEX1 drives transcription of silent VSGs seems equally unlikely, as derepression of silent VSGs is also observed following VEX1-VEX2

double knockdown. VEX1 ChIP-seq reveals strong enrichment of the active VSG but also binding to silent VSGs (Fig. 1a, Supplementary Fig. 1, Supplementary Data File S1, sheet 1), which we believe is more consistent with the homology-dependent VSG-silencing that we propose. We hope that our 'model' (Fig. 6c), which takes account of the new data, helps to clarify.

3.3: It is amazing that these KD cells do not switch. This implies that the ESB is still occupied by the VSG2 ES, assuming that there is only one ESB pol I. If a silent BES occupies the ESB pol I, this should induce in situ switching, therefore VSG2 protein level should be reduced greatly, which was not the case from Figure 2d. However, Pinger et al (2017 Nat Comm) showed that after switching, old VSG half-life is about 5 hours with regular growth rate taken into account. As VEX2 KD and double KD cells are not dividing well, it is possible that VSG2 half-life could be even longer in switched cells. So, although these cells have ESB occupied by a silent ES(s), VSG2 is a major surface protein. I wonder whether relocalization of VEX proteins is a transition step during in-situ switching. If KD cells have cell cycle arrest, they may not complete the switching process (?). Do any of KD cells have cell cycle defects? Have you examined longer time points? If so, any switchers observed?

R3.3: The integrity of the ESB is disrupted after VEX2 knockdown (Fig. 2k, Supplementary Fig. 4c; see above). We've included cell cycle data during VEX2 knockdown, which shows no major perturbation (Fig. S4c). We've also included data over a 10-day time-course following VEX1 knockdown, which shows no switching (Fig. S6b).

3.4: VSG switching seems to be semi-hierarchical. Is there any preference for VSG6 or any of those highly derepressed VSGs during switching? Have you done IF with other VSGs, such as VSG-8 or VSG-17?

R3.4: There is indeed a preference for activation of expression sites with dual promoters. We've added "The same VSG-ESs [containing VSG-6, 8, 15 and 17] were reported to be activated at high rates during VSG switching [Aitchison et al., 2005]". We have also now done IFA with other VSGs - see response R1.5 above.

3.5: It is very interesting that VEX proteins redistribute during transition from BF to PF. Therefore, redistributed VEX proteins may represent a snapshot of normal in-situ switching process. This data also indicates a pol I transcription specific VEX localization, which is more convincing than BMH-21 or Act D treatment, as these drugs cause many other effects that are not related to transcription elongation.

R3.5: We've now added dual VEX1-VEX2 IFA data for PCF (insect-stage) cells (Fig. 4d, inset); also see 1.7 above. We feel that these data are complementary with the inhibitor studies, indicating; "both VEX1 and VEX2 associate with the VSG transcription compartment in a pol-I transcription-dependent and life cycle stage-specific manner". We were also concerned that BMH-21 (specific pol-I inhibitor) might cause other effects. This is why we tested a range of concentrations and exposure times and selected 1 μ M for 30 min for the current study (see Fig. 1 legend); this is 3-fold lower concentration and 6-fold reduced exposure time compared to the 3 μ M for 3 h reported previously prior to immunofluorescence analysis [Kerry *et al.*, 2017].

3.6: Protein-protein interaction was shown by IF, which is good because it shows cellular localization. But IP western is more sensitive to study/confirm CAF1-VEX1 interaction, especially because CAF-1 signal is all over the nucleus. Information of pol I-VEX2 interaction in wild type and VEX1 KD cells would be informative as well.

R3.6: We have added data confirming interaction between the VEX-complex and CAF-1, using IP and protein blotting (Fig. 5c; also see 2.3 above and 3.16 below). We note that four pull-down experiments also support protein-protein interaction

between VEX1, VEX2 and CAF-1 (Fig. 1b, Supp. Fig. 1a-b). We see no evidence for a direct interaction between the VEX-complex and pol-I (see 1.1 above).

3.7: It is interesting that highly derepressed VSGs are different in CAF-1 KD cells (strong derepression of metacyclic VSGs), because both CAF-1 and VEX2 RNAi induce VEX1 redistribution. This needs some explanation. Metacyclic VSGs are more accessible to the ESB or redistributed VEX1 prefers metacyclic VSGs in CAF-1 KD? Or something else. Does CAF-1 affect switching rate? CAF-1 depletion may have differential effects on chromatin structure of ES or metacyclic VSG promoters.

R3.7: We've added "promoter-proximal genes may be particularly susceptible to derepression due to a genome-wide histone chaperone defect." We have not determined whether CAF-1 affects switching rate as, with current approaches, it will not be possible to distinguish between a canonical histone chaperone associated defect and a VEX-complex associated defect.

3.8: Epigenetic memory and epigenetic inheritance are not interchangeable. VEX1 reloading by CAF-1 is epigenetic inheritance, not memory. Epigenetic memory would be a situation in which, when VEX KD cells are re-expressing VEX proteins, how these cells would remember and re-activate the original VSG2, instead of choosing random VSGs.

R3.8: We replaced "memory" with "inheritance" in the title and abstract (also see R1.3 above).

Minor points

3.9: Line 32: There are several canonical UPF homologues in *T. brucei* but their roles are not clear. Since the study does not really show any UPF1 related findings, UPF1 should not be emphasized too much.

R3.9: The UPF1/VEX2 orthologue in mammals has been implicated in telomeric heterochromatin formation and X-chromosome inactivation (see the third paragraph in the Discussion). We've adjusted this line in the abstract in an attempt to clarify further.

3.10: In Figure 2 d FACS plot, why is VSG2 signal very heterogeneous, ranging from 10^2 - 10^5 ? Have you done similar FACS experiments for double KD and CAF-1? Was there any change with VSG2 signal (e.g., reduction?)

R3.10: We discussed this issue with our FACS facility manager and chose to show the dataset without applying a gate to exclude low-staining cells. We believe that low-staining is due to the way the cells are labelled with antibody in solution rather than on slides. We see no change in VSG-2 signal using this approach but feel that this is due to the relatively limited sensitivity of FACS. The drop in VSG-2 is seen by western blot, and proteomics is superior to the use of antibodies for quantitative analyses, hence our use of these approaches with the double KD and CAF-1 strains.

3.11: Line 133: "VSG-6 appeared to accumulate in intracellular compartments as well as the surface" But VSG proteomics was done with proteins released from the cell-surface, this may underrepresent silent VSG levels and types.

R3.11: That's correct, but we felt that it was important to demonstrate multi-VSG expression at the cell surface.

3.12: Line 169: "we did not anticipate derepression of ESAGs at the active VSG" they were not repressed (they are in the active ES), so it should "increased level" or something.

R3.12: We have changed "derepression" to "increased expression".

3.13: Line 198, 317: "completely fail, complete failure" it is not clear what is

“complete” failure. One may think that “complete” failure of allelic exclusion would be expression of silent VSGs in all cells “without having any major VSG”. In KD cells, VSG2 still seems to be the major surface coat. FACS plot would be nice to visualize VSG2 signal loss in VEX double KD cells.

R3.13: We’ve adjusted the terminology to “collapse(s)”. Unfortunately, FACS is not suitably sensitive to demonstrate VSG2 signal loss (see 3.10 above).

3.14: Line 221, 282: ‘positive feedback’ it is not clear what this means. What exactly is feeding back to what?

R3.14: We’ve added the term “sequestration-transcription” to clarify our meaning.

3.15: Figure 4D: this is a better assay to show transcription dependent VEX2 association with ESB, as during BF to PF transition, transcription of the active ES is lost, without compromising other things, unlike drug treatment. Dispersed VEX may capture the next one to be activated, in this case procyclin promoter. You may want to put Figure 1f-1g in supplementary or together with Figure 4.

R3.15: See 3.5 above.

3.16: Line 234-236: To confirm the interaction between CAF-1 and VEX proteins, one can simply do IP western. Tagged proteins and antibodies are available, so this should be simple.

R3.16: See 3.6 above.

3.17: Line 240: although VEX1 redistributed in CAF-1 KD, increase in VEX2 is not due to VEX1 redistribution, because VEX1 RNAi does not reduced VEX2 level. VEX2 level seems to be controlled independently by CAF-1 and VEX1.

R3.17: This is a good point that VEX1 RNAi does not change VEX2 levels. We’ve now stated, “indicating that CAF-1 limits VEX2 abundance, an effect that may be enhanced by excess VEX1 (see above)”.

3.18: Line 283: “VEX-complex... likely important for both establishing and maintaining allelic exclusion” data here is supporting maintenance of exclusion, but not how it is “established”.

R3.18: We’ve adjusted the text here to “important for maintaining, and we speculate, also establishing, allelic exclusion.”

3.19: Line 313: “indeed, a cohesion-dependent delay in active VSG sister chromatid separation facilitates epigenetic inheritance, entirely consistent with a VEX complex reloading step” Derepression of VSGs in cohesion KD was due to the increased rate of in-situ switching, therefore it is not relevant to VEX work.

R3.19: We’ve adjusted the text to clarify our view here; “Thus, premature segregation of sister chromatids following cohesin depletion may compromise VEX-complex reloading and present opportunities for activation of other VSG-ESs.”

Reviewers' Comments:

Reviewer #1:

Remarks to the Author:

The authors have addressed most of my concerns, but not completely (see my comments on specific new Figures below). I agree with reviewer 3 that this manuscript is very dense, the figures are extremely crowded and it will not be easily understood by non VSG aficionados. The model (Fig 6C) is a good idea, but I think it is not very effective because it is too simple.

VEX1-ChIP experiment (Figure 1a)

- The new Figure 1a + Supplementary Figure 1 show the results of VEX1 ChIP. Given that the VSG-containing subtelomeric loci (bloodstream expression sites, BESs) are very similar in sequence, how were the authors able to align and assign reads to individual BESs? I would expect most reads originating from a BES to be non-unique because its sequence is identical to another BES. The authors should explain in simple words in the Result section how the bioinformatic analysis was performed such that almost the entire BES is "covered" by low level of VEX1.

- A visual examination of the mapping of VEX1 in all BESs (supplementary Figure 1) indicates that VEX1 is very abundant at the VSG-telomere boundary of not only BES1 (active BES), but also BES1 and BES4. This is not consistent with the conclusion in the text "An examination of the hemizygous subtelomeric VSG-ESs 6, 17 revealed VSG-2, the active VSG, as the most enriched gene " This needs to be discussed.

Figure 3C (IFA with antibodies against different VSGs) is clear and it nicely shows that a fraction of parasites simultaneously express at least three different VSGs (VSG2, VSG3 and VSG6).

Supplementary Fig 6A is not so clear: it shows quantifications of VSG2/VSG13 cells, but Supplementary Fig 6B only shows VSG2/VSG6.

Figure 4D shows partial co-localization of VEX1 and VEX2 in BSF that were induced to differentiate into PCF for 24hr. I presume, these young "24hr-PCF" parasites are not identical the well-established and "older" PCF cell-line used to do proteomic studies. What is the localization of the two proteins in the parasites used for the proteomic study?

Reviewer #2:

None

Reviewer #3:

Remarks to the Author:

Most of my main concerns have been nicely addressed with additional experiments, VEX1 ChIP seq and Pol I localization in VEX2 KD cells, which provided clearer understanding on subnuclear compartments (VEX1/2, VEX2/ESB, and VEX1) and their (potential) roles. Although it would be nicer to have VEX1 ChIP seq information in VEX2 KD cells, for example, to see that redistributed VEX1 no longer binds silent VSGs and also VEX2 ChIP seq as well (to see where VEX2 might lie within ES?), but these can be done in future studies. I am satisfied with revised manuscript.

Their homology-dependent VSG silencing model has a stronger support with new data; VEX1 associates with the active VSG region within ES but also with silent VSGs (though with lower affinity: this is interesting). I wonder whether having silent VSGs near the active ES can facilitate the 'homology-mediated silent VSG repression'. VEX2 may bridge Pol I and VEX1-VSGs, and controls ES body integrity as well as VEX1 function (therefore, VEX2 KD cells show the strongest derepression of

silent VSGs).

VEX2 seems to have roles outside of VSG expression, as nucleolar Pol I IFA signal, not just ES-Pol I, disappeared in VEX2 knockdown cells. This may explain why VEX2 is essential but not VEX1. It is puzzling to me, what is driving transcription of silent VSGs in VEX mutant cells. Or could it be because silent VSG RNAs are more stable in KD cells (post-transcriptional gene silencing, a type of homology-dep gene silencing used in plant)? These can be addressed further in another study.

Two minor comments are below

3.14: Line 221, 282: 'positive feedback' it is not clear what this means. What exactly is feeding back to what?

R3.14: We've added the term "sequestration-transcription" to clarify our meaning.

Comment on R3.14: Wikipedia defines 'positive feedback', A produces more of B which in turn produces more of A. I am not sure if data showed any 'feedback'. From data, it seems more like co-dependent relationship between sequestration and transcription, rather than positive feedback loop.

3.19: Line 313: "indeed, a cohesion-dependent delay in active VSG sister chromatid separation facilitates epigenetic inheritance, entirely consistent with a VEX complex reloading step" Derepression of VSGs in cohesion KD was due to the increased rate of in-situ switching, therefore it is not relevant to VEX work.

R3.19: We've adjusted the text to clarify our view here; "Thus, premature segregation of sister chromatids following cohesin depletion may compromise VEX-complex reloading and present opportunities for activation of other VSG-ESs."

Comment on R3.19: it is still not clear as there is no evidence. It is possible that VEX-cohesin may have some interaction. But compromised VEX complex reloading should impact VSG silencing, not switching, as author mentioned that even after 10 days of KD, no switchers have been detected (if KD would work that long). I am not sure if cohesion part is really relevant for this work. But this is a minor comment.

NCOMMS-18-32858A

"Monoallelic expression and epigenetic inheritance sustained by a *Trypanosoma brucei* variant surface glycoprotein exclusion complex"

Reviewer 1:

The authors have addressed most of my concerns, but not completely (see my comments on specific new Figures below). I agree with reviewer 3 that this manuscript is very dense, the figures are extremely crowded and it will not be easily understood by non VSG aficionados. The model (Fig 6C) is a good idea, but I think it is not very effective because it is too simple.

We were happy to hear that the model helps. This has now been extended and described in more detail in the Figure legend to further clarify our thinking.

VEX1-ChIP experiment (Figure 1a)

-The new Figure 1a + Supplementary Figure 1 show the results of VEX1 ChIP. Given that the VSG-containing subtelomeric loci (bloodstream expression sites, BESs) are very similar in sequence, how were the authors able to align and assign reads to individual BESs? I would expect most reads originating from a BES to be non-unique because its sequence is identical to another BES. The authors should explain in simple words in the Result section how the bioinformatic analysis was performed such that almost the entire BES is "covered" by low level of VEX1.

- A visual examination of the mapping of VEX1 in all BESs (supplementary Figure 1) indicates that VEX1 is very abundant at the VSG-telomere boundary of not only BES1 (active BES), but also BES1 and BES4. This is not consistent with the conclusion in the text "An examination of the hemizygous subtelomeric VSG-ESs 6, 17 revealed VSG-2, the active VSG, as the most enriched gene " This needs to be discussed.

We've now commented on uniqueness filtering in the Results section. We've also adjusted the text in this section to clarify that the VSG-2 'protein coding sequence' was the most enriched.

Figure 3C (IFA with antibodies against different VGS) is clear and it nicely shows that a fraction of parasites simultaneously express at least three different VSGs (VSG2, VSG3 and VSG6). Supplementary Fig 6A is not so clear: it shows quantifications of VSG2/VSG13 cells, but Supplementary Fig 6B only shows VSG2/VSG6.

Supp. Fig. 6A, as well as the relevant section in the main text and the legend, have now been adjusted to clarify.

Figure 4D shows partial co-localization of VEX1 and VEX2 in BSF that were induced to differentiate into PCF for 24hr. I presume, these young "24hr-PCF" parasites are not identical the well-established and "older" PCF cell-line used to do proteomic studies. What is the localization of the two proteins in the parasites used for the proteomic study?

That is correct. The proteomic studies were carried out with established insect-stage cells prior to working with bloodstream form cells as, although as far as we are aware allelic exclusion does not operate, cryomilling protocols were established in these insect-stage cells. Understanding the role and behavior of the VEX-complex in insect-stage cells is certainly of interest but this will, in our view, require further study, likely including knockdown and RNA-seq and also ChIP-seq.

Reviewer 3:

Most of my main concerns have been nicely addressed with additional experiments, VEX1 ChIP seq and Pol I localization in VEX2 KD cells, which provided clearer understanding on subnuclear compartments (VEX1/2, VEX2/ESB, and VEX1) and their (potential) roles. Although it would be nicer to have VEX1 ChIP seq information

in VEX2 KD cells, for example, to see that redistributed VEX1 no longer binds silent VSGs and also VEX2 ChIP seq as well (to see where VEX2 might lie within ES?), but these can be done in future studies. I am satisfied with revised manuscript.

We do indeed intend to carry out further ChIP-seq experiments as part of our future studies.

Their homology-dependent VSG silencing model has a stronger support with new data; VEX1 associates with the active VSG region within ES but also with silent VSGs (though with lower affinity: this is interesting). I wonder whether having silent VSGs near the active ES can facilitate the 'homology-mediated silent VSG repression'. VEX2 may bridge Pol I and VEX1-VSGs, and controls ES body integrity as well as VEX1 function (therefore, VEX2 KD cells show the strongest derepression of silent VSGs).

We wonder whether silent telomeres are clustered with the active telomere but this is rather speculative at this stage. In relation to the second point, we do not see evidence for bridging to pol-I since "no pol-I components were enriched" along with the VEX-complex.

VEX2 seems to have roles outside of VSG expression, as nucleolar Pol I IFA signal, not just ES-Pol I, disappeared in VEX2 knockdown cells. This may explain why VEX2 is essential but not VEX1. It is puzzling to me, what is driving transcription of silent VSGs in VEX mutant cells. Or could it be because silent VSG RNAs are more stable in KD cells (post-transcriptional gene silencing, a type of homology-dep gene silencing used in plant)? These can be addressed further in another study.

We do not believe that VEX directly impacts the nucleolus nor that this is post-transcriptional silencing. We state "derepressed VSG-ESs deplete the nucleolus and the ESB of pol-I, redistributing the polymerase to multiple extranucleolar sites". i.e. loss of pol-I from the nucleolus appears to be an indirect consequence of VSG-ESs derepression. We've added a comment and citation (Kassem *et al.*, 2014) to the Introduction to clarify silent VSGs are not transcribed. We also hope that the extended model (Fig. 6C) helps in this regard.

Two minor comments are below

3.14: Line 221, 282: 'positive feedback' it is not clear what this means. What exactly is feeding back to what?

R3.14: We've added the term "sequestration-transcription" to clarify our meaning.

Comment on R3.14: Wikipedia defines 'positive feedback', A produces more of B which in turn produces more of A. I am not sure if data showed any 'feedback'. From data, it seems more like co-dependent relationship between sequestration and transcription, rather than positive feedback loop.

We've removed reference to 'positive feedback' from the Results section and now only raise this possibility in the Discussion. We do indeed believe that VEX-complex sequestration produces more transcription, which in turn produces more sequestration.

3.19: Line 313: "indeed, a cohesion-dependent delay in active VSG sister chromatid separation facilitates epigenetic inheritance, entirely consistent with a VEX complex-reloading step" Derepression of VSGs in cohesion KD was due to the increased rate of in-situ switching, therefore it is not relevant to VEX work.

R3.19: We've adjusted the text to clarify our view here; "Thus, premature segregation of sister chromatids following cohesin depletion may compromise VEX-complex-reloading and present opportunities for activation of other VSG-ESs."

Comment on R3.19: it is still not clear as there is no evidence. It is possible that VEX-cohesin may have some interaction. But compromised VEX complex reloading should impact VSG silencing, not switching, as author mentioned that even after 10

days of KD, no switchers have been detected (if KD would work that long). I am not sure if cohesion part is really relevant for this work. But this is a minor comment. We're suggesting that VEX1 would be liberated during cohesin knockdown. This may present opportunities for other sites to compete for binding to / recruitment of this factor. VEX1 knockdown would not be expected to present similar opportunities. We've further adjusted this text to clarify.